# Robotic multi-probe single-actuator inchworm neural microdrive

Richard D Smith*, Ilya Kolb, Shinsuke Tanaka, Albert K Lee, Timothy D Harris, Mladen Barbic*

Janelia Research Campus, Howard Hughes Medical Institute, Ashburn, United States

**Abstract** A wide range of techniques in neuroscience involve placing individual probes at precise locations in the brain. However, large-scale measurement and manipulation of the brain using such methods have been severely limited by the inability to miniaturize systems for probe positioning. Here, we present a fundamentally new, remote-controlled micropositioning approach composed of novel phase-change material-filled resistive heater micro-grippers arranged in an inchworm motor configuration. The microscopic dimensions, stability, gentle gripping action, individual electronic control, and high packing density of the grippers allow micrometer-precision independent positioning of many arbitrarily shaped probes using a single piezo actuator. This multi-probe single-actuator design significantly reduces the size and weight and allows for potential automation of microdrives. We demonstrate accurate placement of multiple electrodes into the rat hippocampus in vivo in acute and chronic preparations. Our robotic microdrive technology should therefore enable the scaling up of many types of multi-probe applications in neuroscience and other fields.

## Editor's evaluation

This work describes a new device for controlling the positioning of chronically-implanted movable electrodes in the brain. Potentially replacing microscrew-based devices with a cunningly-engineered electromechanical system, this highly accurate yet low-cost alternative could be an important milestone for systems neuroscientists currently using microdrives. While the design and demonstration of the system are solid and generated significant excitement, the limited demonstration of the robustness of the device in long term many electrode configurations was perhaps incomplete. On the whole, this promising study suggests that methodological advances may yet revolutionize neuroscience utilizing arrays of movable electrodes.

*For correspondence:
rdsmith@caltech.edu (RDS);
Mladen.Barbic@nyulangone.org
(MB)

## Introduction

The miniaturization of mechanical, electrical, optical, and other devices has been steadily ongoing for decades, enabled by advances in manufacturing, electronics, and novel mechanical designs. Miniaturization is motivated by demands for enhancing sensitivity and functionality, reducing costs, parallelizing and scaling of experiments, and reducing weight and size to enable operation in constrained spaces. Aided by such technological advances, tools and techniques in neuroscience have undergone tremendous progress, enabling the study of the living, behaving brain (*Wise et al., 1970*; *Wise and Najafi, 1991*; *Kralik et al., 2001*; *Wise et al., 2004*; *Buzsáki, 2004*; *Wise et al., 2008*; *Harrison, 2008*).

To perform in vivo measurements or perturbations in acutely or chronically implanted animals, a variety of electrical, chemical, and optical probes are mechanically inserted into the brain (*McNaughton et al., 1983*; *Gray et al., 1995*; *Jog et al., 2002*; *Nicolelis et al., 2003*; *Csicsvari et al., 2003*; *Blanche et al., 2005*; *Cogan, 2008*; *Royer et al., 2010*; *Du et al., 2011*; *Szuts et al.,*

*2011*; *Andrásfalvy et al., 2014*; *Berényi et al., 2014*; *Schwarz et al., 2014*; *Canales et al., 2015*; *Jun et al., 2017*; *Feiner and Dvir, 2018*; *Hunt et al., 2019*). Since the brain is anatomically structured into different functional regions, these probes must be precisely placed in the region of scientific or clinical interest, often with micron-scale accuracy. A number of microfabricated electrodes have enabled recording from neurons localized to a single plane (*Maynard et al., 1997*) or column (*Jun et al., 2017*), but for many applications multiple independently actuated electrodes are required to record neural activity in a geometry-flexible and post-implantation adjustable fashion. Flexible-sheet array implants (*Lu et al., 2016*) can record over a large area but are typically only applicable to the external surface as in cortical experiments. Shank-style probes are often limited in the number and density of electrodes along their length, and therefore generally require alignment during and after implantation due to tissue movement, local scarring, or cell loss. While highly integrated probes (*Jun et al., 2017*; *Steinmetz et al., 2021*) with dense electrodes along a larger portion of the shank can obviate the need for adjustment, there are currently limitations to the three-dimensional target geometries that are realizable with these probes. Furthermore, beyond the standard electrode types available, such probes do not exist for the wide variety of other types of electrodes or measurement/ manipulation devices that have been developed (*Royer et al., 2010*; *Canales et al., 2015*; *Hunt et al., 2019*). For these reasons, the ability to continually and independently position multiple probes remains important within experimental neuroscience.

Therefore, a mechanical positioner, commonly termed a microdrive, is usually employed for such placement tasks, and its design has also undergone steady technological development and improvement over the years (*Humphrey, 1970*; *Ainsworth and O'Keefe, 1977*; *Krüger, 1983*; *Kubie, 1984*; *Korshunov, 1995*; *Nichols et al., 1998*; *Venkatachalam et al., 1999*; *Vos et al., 1999*; *Szabó et al., 2001*; *Keating and Gerstein, 2002*; *Jeantet and Cho, 2003*; *Swadlow et al., 2005*; *Korshunov, 2006*; *Lansink et al., 2007*; *Tóth et al., 2007*; *Battaglia et al., 2009*; *Haiss et al., 2010*; *Galashan et al., 2011*; *Santos et al., 2012*; *Voigts et al., 2013*; *Liang et al., 2017*). However, presently available microdrives have features that limit the extent to which the number and density of probes can be scaled up. Microdrives tend to be manually operated devices where each independent probe is assigned its own lead-screw-based positioner. Therefore, increasing the number of neural probes mounted in such drives often makes them complex, bulky, and heavy and also makes the practice of probe placement increasingly cumbersome and time-consuming. Work has gone into developing motorized microdrive devices that would allow for more efficient electrophysiological data collection through parallelization and remote computer-controlled operation without human intervention (*Reitboeck, 1983*; *Eckhorn and Thomas, 1993*; *Fee and Leonardo, 2001*; *Johnson and Welsh, 2003*; *Cham et al., 2005*; *Venkateswaran et al., 2005*; *Gray et al., 2007*; *Sato et al., 2007*; *Yamamoto and Wilson, 2008*; *Park et al., 2008*; *Jackson et al., 2010*; *Yang et al., 2011*; *Kodandaramaiah et al., 2012*; *Zoll et al., 2019*). Yet, in these designs too, each independent neural probe is assigned its own motorized positioner, which often makes the size, weight, complexity, and expense of the microdrive prohibitive and limits their ability to scale to large numbers of probes.

Here, we present a fundamentally different approach to microdrive design that directly addresses the limitations of previous approaches. First, we developed a novel, reusable, electronically controlled, densely packable, phase-change-based microgripper for holding and releasing probes of arbitrary shapes. Then we arranged arrays of microgrippers into an inchworm motor configuration so that a single piezo actuator can independently translate many probes over an unlimited range of travel. This multi-probe single-actuator (MPSA) concept therefore allows for a significant reduction in microdrive complexity, size, weight, and cost, while still providing micron-scale independent positioning of each neural probe. As a proof of concept, we constructed and tested the operation of this microdrive while loaded with classic twisted wire tetrodes – a widely used type of neural electrode. We performed remote-controlled, independent placement of multiple tetrodes with micrometer precision into the CA1 region of the rat hippocampus in vivo in acute and chronic settings. We have also demonstrated the versatility of our approach by independently adjusting an array of closely packed glass micropipettes. Our method should therefore allow large-scale monitoring and manipulation of activity across the brain during behavior using a wide variety of probes.

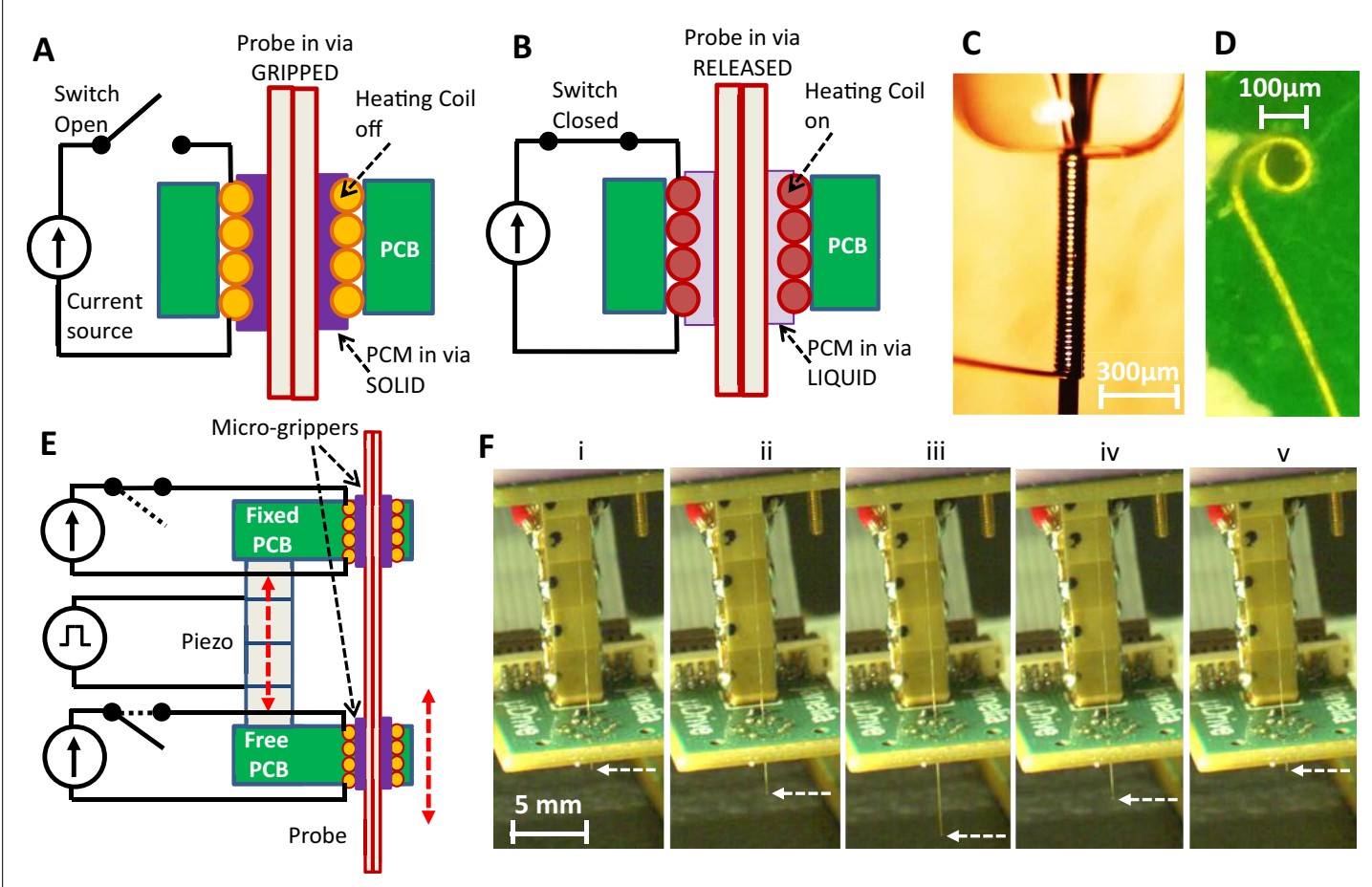

**Figure 1.** Microgripper and single-probe inchworm microdrive design. (**A–B**) Side view of microgripper design. A resistive heating coil is installed in a printed circuit board (PCB) via. A probe is placed through the coil bore, and the bore is filled with a temperature-responsive phase change material (PCM). With the heating coil off (**A**), PCM is in the solid state, and the probe is 'gripped'. When the coil is on (**B**), PCM is in the liquid state, and the probe in the microgripper is 'released' and can be moved axially through the bore. The PCM can be maintained in the microgripper through capillary action and does not exit the bore. (**C**) Photograph of the fabricated heating coil before it is placed into the board. (**D**) Photograph of the top view of the heating coil installed into the PCB. Schematic (**E**) and photos (**F**) of the inchworm motor. Two parallel PCBs, each with the installed microgrippers, are aligned along the probe axis and joined by a single piezo actuator. The top board is fixed while the bottom board is movable by the motion of the voltage-controlled piezo actuator. Sequential electronically controlled probe gripping and piezo extension and contraction (described in *Figure 2*) are used to translate the probe (white dashed arrow points to the probe tip) in either direction (up or down), as shown in F(i)–(v) and *Video 1*.

## Results

### Phase change material-filled resistive heater coil microgripper

We introduce the new probe microgripper in *Figure 1A–B*, which schematically describes the functional design of the device. A helical resistive wire coil embedded in a printed circuit board (PCB) non-plated via is connected such that an electric current from a power source can be passed through the coil. This allows for the electronically controlled change of the gripper's temperature through Joule heating. A probe is threaded through the bore of the coil, and the remainder is filled with a temperature controlled phase change material (PCM) (*Barbic et al., 2017*). When the heater coil does not carry current (*Figure 1A*), the gripper is at the ambient temperature, the PCM in the bore is in the solid state, and therefore the neural probe is 'gripped' (*Barbic et al., 2017*). When the heater coil carries current (*Figure 1B*), the gripper heats up, the PCM in the bore melts and goes into the liquid state. The neural probe is therefore 'released' (*Barbic et al., 2017*) and can be moved axially through the gripper. Critically, due to the microscopic dimensions of the space between the gripper and the neural probe, the PCM can be maintained in the bore through capillary action without exiting during probe motion. *Figure 1C* shows a photograph of the side view of the fabricated helical resistive

microheater before it is placed into a PCB, while *Figure 1D* shows a photograph of the top view of the heater coil after it is installed (see Materials and methods).

We emphasize several important features of this probe gripper design. First, the gripper is operated electronically which allows for remote control. Second, the gripper is dimensionally microscopic (~75 μm inner diameter/110 μm outer diameter, *Figure 1C–D*) and of similar cross-sectional size as the neural probe that it grips (twisted wire tetrode probe ~55 μm diameter). Therefore, the gripper takes up minimal PCB area, and many probes can be densely packed, as we describe in a later section. Third, the probe gripped within the bore of the device can be of any cross-sectional shape, as the liquid PCM in the released state conforms to the probe shape before it solidifies into the gripped state. This presents an opportunity to use the device to grip a variety of probes such as electrical testing probes, optical fibers, silicon neural electrodes, glass pipettes/capillary probes, carbon fibers, and ultrasonic probes. For the purposes of demonstration, in addition to neural twisted wire tetrode electrodes, we also demonstrate in a later section gripping and translation of glass micropipettes.

## Single-probe inchworm microdrive operation

The above described neural probe microgripper design is integrated into an inchworm motor structure, as shown diagrammatically in *Figure 1E*. The motor is constructed from two parallel PCBs, each with the embedded helical coil heater grippers aligned along the neural probe axis. The two parallel boards are joined by a single piezoelectric block stack actuator that is controlled by a piezo actuator voltage driver (see Materials and methods). In this diagram, the top board is fixed,while the bottom board is free and movable by the motion of the piezo actuator. The functional concept will remain the same if the roles are reversed so that the choice can be dictated by application considerations. The inchworm motor steps (*Devasia et al., 2007*; *Ouyang et al., 2008*) of sequential probe gripping and piezo extension and contraction (as we describe in detail in the next section) are used for the neural probe translation shown in *Figure 1F*. By electronically controlling the current through the resistive heaters (and therefore the gripping and releasing of the probe in the bores) in the respective top and bottom boards (*Figure 1E*), the neural probe in the device can be translated in either direction (up or down) with the piezo actuator, as shown in *Figure 1F* and *Video 1*.

We note several important features of this inchworm motor design. First, the piezo actuator is, just like the probe grippers, controlled electronically which allows for remote control. Second, the piezo actuator has micron-scale translation capability, accuracy, and repeatability. This is critical when precise positioning of the neural electrodes is required for targeted neural recordings. Third, the inchworm motor design in general allows for the long range forward or backward motion of the probe with indefinite translation range. This is because the piezo can make a practically unlimited number of microscopic steps, as *Figure 1F* illustrates.

We now provide detailed description, in *Figure 2*, of the sequence of electronic actuation signals sent to the inchworm microdrive for its three distinct probe translation operations: (a) downward translation step, (b) upward translation step, and (c) no translation step. Translation of a probe is accomplished using a precisely timed sequence of electronic actuation steps (*Figure 2*). The sequence to move a probe one step downward is depicted in *Figure 2A*. In the first step of the sequence, *Figure 2A* (i), the top heating coil is turned on by passing a controlled current through it. This causes the temperature of the top microgripper to rise, such that the PCM in the top heater bore melts and releases the probe. While the top

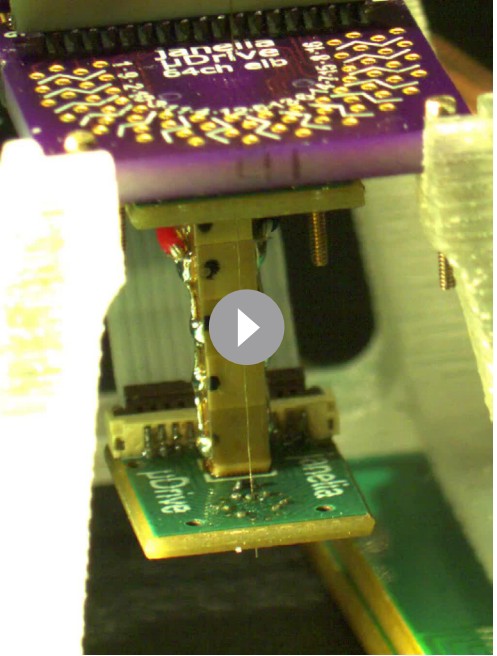

**Video 1.** Up and down translation of a single tetrode in the robotic inchworm microdrive.
https://elifesciences.org/articles/71876/figures#video1

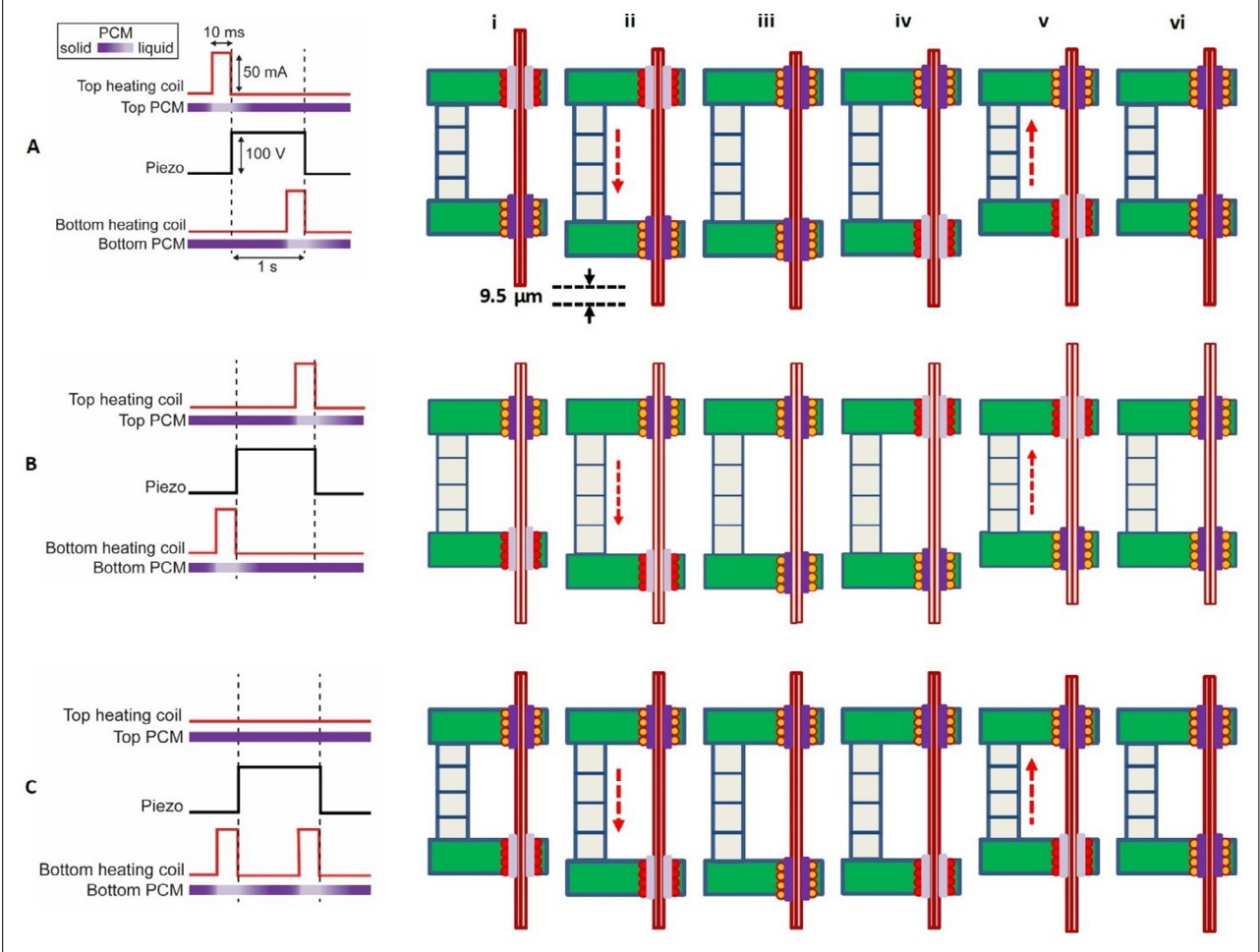

**Figure 2.** Sequence of electronic actuation steps used by the microdrive for a single translation step. (**A**) A single downward step (here 9.5 µm) of the probe is performed by six inchworm motor phases (i)–(vi). Long-distance downward probe translation can be accomplished through an unlimited number of repeated single step sequences. In this schematic, the top gripper board is fixed, while the bottom gripper board moves with the piezo extension or contraction. Positive voltage applied to the piezo causes it to extend. (**B**) A sequence change in the activation of the microgrippers results in a single upward step. Importantly, the actuation signal to the piezo remains the same. (**C**) Single no-step sequence, applied to those probes that are to remain in place while other probes are moved in a multi-probe microdrive. The piezo is still actuated, extending and contracting as in (**A**) and (**B**), but the sequence of the actuation signals to the top and bottom board heaters is such that the probe does not move. Pulse timing is not shown to scale.

microgripper's PCM is liquid, a voltage is applied to the piezo actuator, causing it to extend downward, as shown in *Figure 2A* (ii). Since the probe is still gripped in the bottom board, which moves with the piezo actuator, the probe is also moved downward the same distance, sliding through the top microgripper. After the top heater is turned off, the top PCM remains liquid for a short time while the heat diffuses away. Subsequently, the PCM solidifies and the top microgripper re-grips the probe while, importantly, the piezo actuator is still extended, as shown in *Figure 2A* (iii). Next, the bottom heating coil is turned on, which causes the bottom microgripper to release the probe, as shown in *Figure 2A* (iv). While the bottom PCM is liquid, the voltage applied to the piezo actuator is set to zero, causing the piezo to contract upward, back to its initial length, as shown in *Figure 2A* (v). Since the probe is released in the bottom board (which moves with the piezo actuator) and is still gripped in the stationary top board, the probe remains fixed and does not move back with the piezo actuator, as shown in *Figure 2A* (v). After a period with the bottom heater off, the bottom PCM solidifies, and the bottom microgripper re-grips the probe, as shown in *Figure 2A* (vi). The result of this sequence of six specifically timed steps is a single downward translation step of the probe with a step size controlled by the voltage applied to the piezo actuator.

A simple change to the sequence of heater and piezo actuations reverses the translation of the probe. That is, if the bottom microgripper is released before the piezo extension, and the top microgripper before the contraction, then the motion is reversed, yielding a single upward translation step (*Figure 2B*). Finally, if only the bottom microgripper is released before each piezo actuation, then the probe will be decoupled from the piezo actuation, and there will be no translation step of the probe (*Figure 2C*). The ability to keep probes stationary as the piezo moves is critical to the MPSA microdrive concept, as described in the next section.

## Multi-probe single-actuator inchworm microdrive operation

The availability of the three distinct probe translation operations of the microdrive while the piezo actuator goes through the same extension/contraction steps (as described in *Figure 2*) allows the microdrive to independently translate multiple closely packed, parallel probes while still only using a single piezo actuator. This is accomplished by the placement of multiple microgrippers into the top and bottom PCBs, such that they are in close proximity and independently electronically controllable for the necessary gripping actuation steps described in *Figure 2*. The schematic of the MPSA inchworm motor microdrive is shown in *Figure 3A*, while a photograph of the side view of a model device with three loaded twisted wire tetrode electrodes is shown in *Figure 3B*. Here, the top gripper board is fixed, while a single piezo actuator moves the bottom gripper board. Independent current sources are connected to each of the helical coil heaters, while hardware and software developed for the microdrive direct the sequence of operations performed by the microdrive's electronic components (independent top and bottom board helical coil heaters and the piezo actuator). A schematic of the microdrive electronic circuitry is shown in *Figure 3C*. We have prepared neural robotic microdrives with up to 16 independent helical coil heater microgrippers in each board, typically in a hexagonal closely packed array, with 300 µm center-to-center microgripper spacing, as shown in *Figure 3D–F* (see Materials and methods). It is important to note that the size and weight of the microdrive remain essentially the same with the increase in the number of loaded neural probes, as the size and weight of additional probes and grippers are nearly negligible. We also note that in addition to the resistive helical coil shown in *Figure 1C and D*, the resistive heater gripper could be potentially constructed from plated or deposited resistive material in the via, a chip resistor installed adjacent to the via, or other manufacturing methods.

In *Figure 3G* and *Video 2*, we demonstrate the ability of the MPSA microdrive to independently move multiple probes (twisted wire tetrodes). The sequence of signals to the pairs of top and bottom board microgrippers, as described in *Figure 2*, independently and simultaneously controls the motion of each probe (*Figure 3G*).

## Multi-probe single-actuator inchworm microdrive characterization

To independently move the probes, the microgrippers must be sufficiently thermally independent. The close packing of probes for many applications, however, could cause significant thermal coupling and risk unintentional probe release. We therefore determined whether and under what conditions the microgrippers could function in a pattern useful for neural recordings. Simulations and translation testing revealed the importance of the short distance between a microgripper's heater and PCM relative to the distance between microgrippers (which are separated by PCB laminate, a composite typically having low thermal diffusivity) and that short, controlled heat pulses would be critical (see Materials and methods). A hexagonal close packed array of 16 tetrode-sized microgrippers, with 300 µm center-to-center spacing was accomplished when the heating coils (~125 Ω) were operated with ~45 mA, yielding a~250 mW heat output and release of the docosane-filled microgrippers in 4–12.5 ms depending on their initial temperature. Heating for longer than 12.5 ms from room temperature was functionally unnecessary and risked unintentional release of adjacent grippers. We additionally increased the board cooling and therefore cycle rate by adding internal copper layers to the gripper boards outside of the microgripper region. Step rates of 0.5 Hz (~300 µm/min) could be continuously maintained with steady-state board warming of a few degrees. Faster step rates of 2.5 Hz (~1.5 mm/min) could be maintained for brief periods of time. Greater timing flexibility and, in particular, much faster step rates were also possible when reduced numbers of probes were used, due to the reduced heat dissipation in the microgripper region.

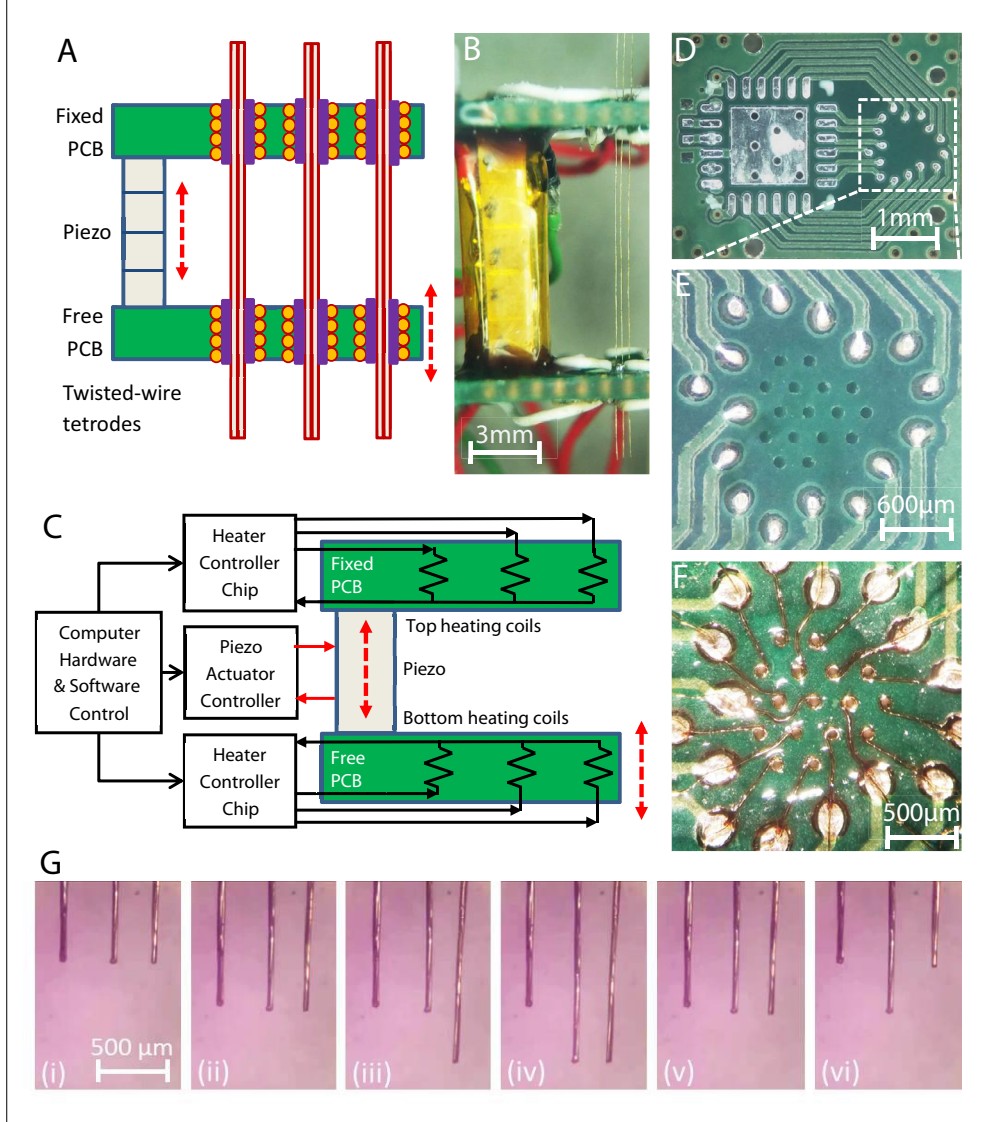

**Figure 3.** Multi-probe translation capability. (**A**) A multi-probe single-actuator (MPSA) microdrive with independently movable probes. Axially aligned pairs of independently controlled microgrippers are integrated into the top and bottom microgripper PCBs. The top board is fixed, and the bottom board is free to be moved by the piezo. (**B**) Side view of a model MPSA microdrive loaded with three tetrodes (right). (**C**) Schematic of the electronic circuitry. Independent current sources are electrically connected to heater coils in the vias, while hardware and software control the sequence of electronic actuation signals. (**D–F**) An example unpopulated printed circuit board (**D**) with 16 drilled vias (**E**) for placement of 16 microgrippers in a hexagonal close packed arrangement (**F**), with 300 µm center-to-center spacing. Demonstration of multiple independent probe movements is shown in *Video 2* and (**G**): (i) with all three probes aligned in height, (ii) all three probes moved together downward, (iii) just the right probe is moved downward, while the left and middle are held stationary, (iv) just the middle probe is moved downward, while the left and right are held stationary, (v) the right and middle probes are moved together upward, while the left is held stationary, (vi) the right and left are moved upward, while the middle electrode is held stationary. The piezo actuator always receives the same actuation sequence of signals throughout these different independent electrode translation options (down step, up step, and no step sequences of *Figure 2*). Forty-two piezo steps were taken between each consecutive image.

The mechanical characteristics of the MPSA microdrive were quantified with optical metrology on both tetrodes and pipettes (*Figure 4A*). A tetrode was first moved using the maximum rated voltage for the piezo, producing reliable stepwise motion as expected from the inchworm motor scheme (*Figure 4B*, top). Step size can be decreased for finer resolution motion by reducing the

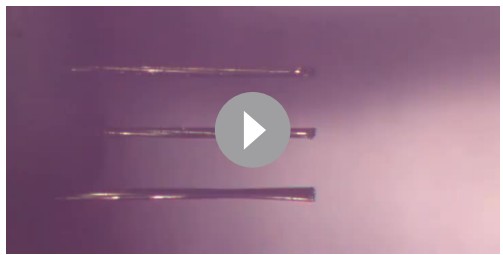

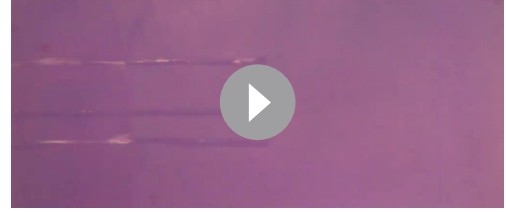

**Video 2.** Translation of three tetrodes by the robotic inchworm microdrive in air, demonstrating independent tetrode positioning control with the multi-probe single-actuator concept.
https://elifesciences.org/articles/71876/figures#video2

**Video 3.** Translation of three tetrodes by the robotic inchworm microdrive demonstrating significant reduction of lateral motion of the tetrodes in agar.
https://elifesciences.org/articles/71876/figures#video3

voltage applied to the piezo during each step (*Figure 4B*, bottom). The displacement of the probe matched the actuation distance of the piezo stack throughout the 100 V range of the driving signal (*Figure 4C*). The minimum reliable step size was measured to be ~600 nm, near the limit of our optical resolving power, and the maximum was 9.5 µm, corresponding to the stroke size of the piezo.

In a separate experiment, we characterized the accuracy and repeatability of probe positioning. Accuracy was defined as the difference between a commanded probe displacement (1 mm and 4 mm) and the actual measured probe displacement. Repeatability was defined as the displacement error associated with approaching a single position from different starting locations. A tetrode loaded into the microdrive and advanced by 9.5 µm steps exhibited a mean on-axis accuracy of 7 µm within a 1 mm total displacement range and 58 µm within a 4 mm total displacement range (*Figure 4D*). Intuitively, this means that if the probe is commanded to move 1 mm, it will overshoot or undershoot by an of average 7 µm. It is expected that accuracy could be increased over a longer travel range with closed loop motion. In a separate experiment, we measured repeatability. The tetrode loaded in the microdrive exhibited a mean repeatability of 38.3 µm (range: 5.5–106.1 µm, n=6 positions). In an agar brain phantom, the repeatability improved to 4.7 µm (range: 0.9–12.9 µm, n=6 positions). A possible cause for the improvement was the reduction of off-axis probe motion in the microgripper when the probe tip is constrained in the agar. Intuitively, this means that if the probe is commanded to move to a specific position many times, the spread of the resulting probe positions will be on average 4.7 µm in agar. In practice, the position of a probe will be optimized using its measured signals or stimulation effects. Therefore, it is expected that this closed loop feedback will minimize the position inaccuracy.

Any coupling in the motion between adjacent probes is undesirable; therefore, in a separate experiment, we measured the effect of moving one pipette on a neighboring stationary pipette positioned 300 µm away (*Figure 4E*). When both pipettes are in air, a single 9.5 µm step of the moving pipette caused the stationary one to move momentarily 3.6±2.1 µm on-axis and 23.8±4.4 µm (n=5 steps)

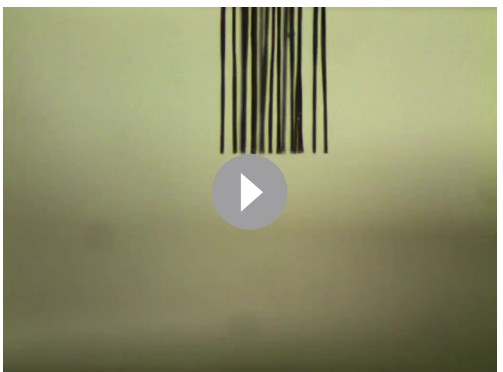

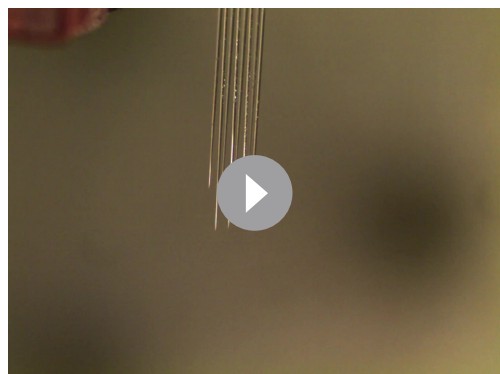

**Video 4.** Translation of 16 independent tetrodes by the robotic inchworm microdrive in air with the multi-probe single-actuator (MPSA) concept.
https://elifesciences.org/articles/71876/figures#video4

**Video 5.** Translation of eight independent glass pipettes by the robotic inchworm microdrive in air with the multi-probe single-actuator concept.
https://elifesciences.org/articles/71876/figures#video5

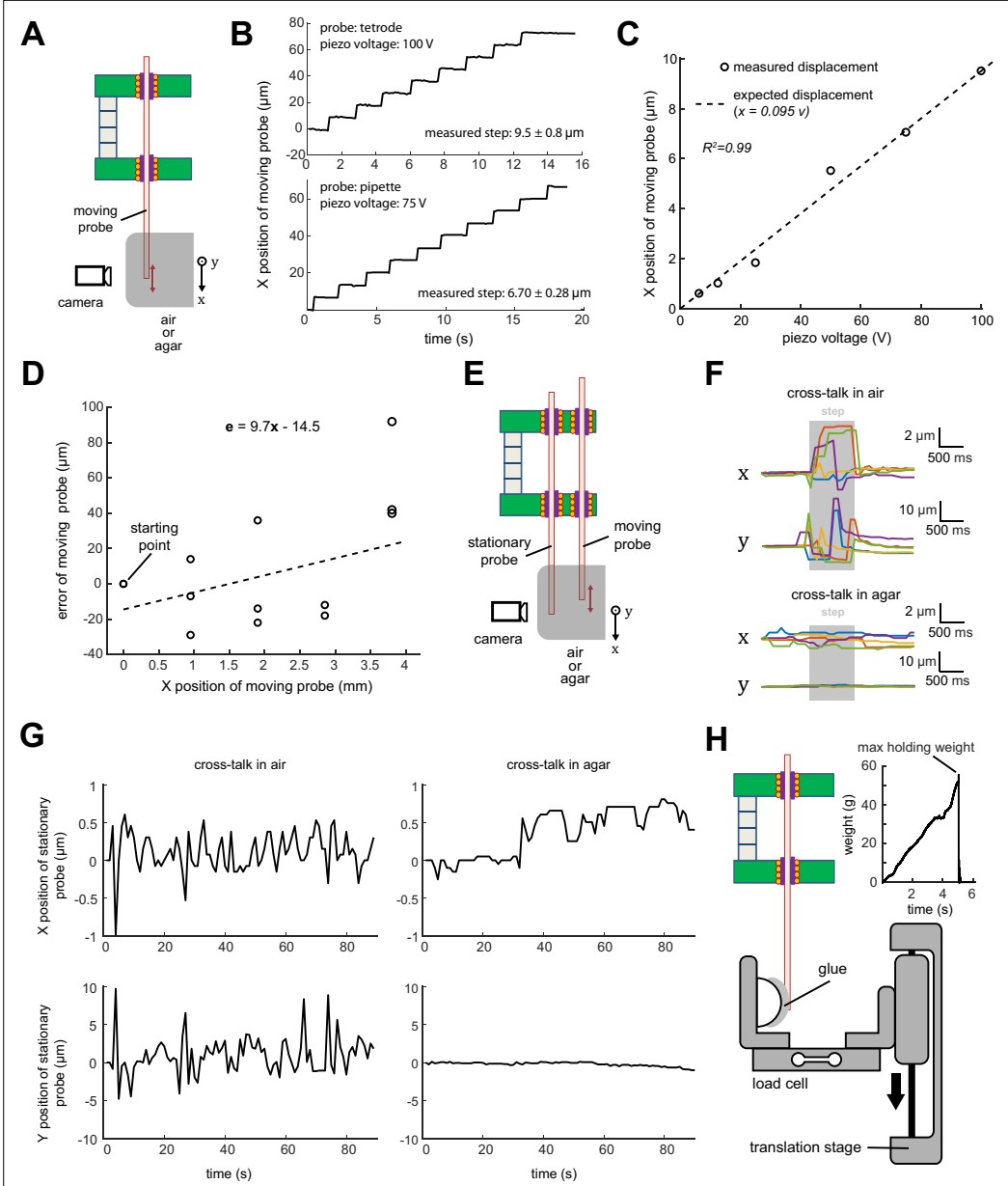

**Figure 4.** Multi-probe single-actuator (MPSA) microdrive mechanical characterization. (**A**) Experimental setup diagram: microdrive is clamped such that probes move in the field of view of the camera or microscope. In some experiments, probe tips were embedded in agar. Motion in the X direction indicates 'on-axis' in the direction of the probe, motion in the Y direction indicates 'off-axis' lateral motion (in and out of the page). (**B**) Top: measured stepwise motion of a tetrode loaded into the microdrive. Step size was set to 9.5 μm (maximum piezo voltage). Bottom: measured stepwise motion of a micropipette loaded into the microdrive. Step size was set to 6.7 μm (75 V piezo voltage). Motion was imaged at two frames per second. (**C**) Correspondence between piezo command voltage and measured tetrode step size. (**D**) Accuracy of probe motion over 4 mm travel range. (**E**) Experimental setup for measuring cross-talk between a moving probe and a stationary neighboring probe. (**F**) Representative displacements in X (on-axis) and Y (off-axis) of a stationary pipette during a single step of a neighboring probe. Pipettes moving in agar brain phantoms (bottom) exhibited smaller displacements than those moving in air (top). Step duration is marked in gray and corresponds to actuation of the piezo stack. N=5 steps are shown. (**G**) Long-term drift of stationary probe during continuous stepping motion of moving probe (9.5 μm step size). (**H**) Experimental setup for measuring maximum holding weight of a single microgripper. A translation stage moved the load cell until the tetrode came loose from the microgripper. The maximum holding weight was considered to be the weight supported by the microgripper immediately before breaking (inset).

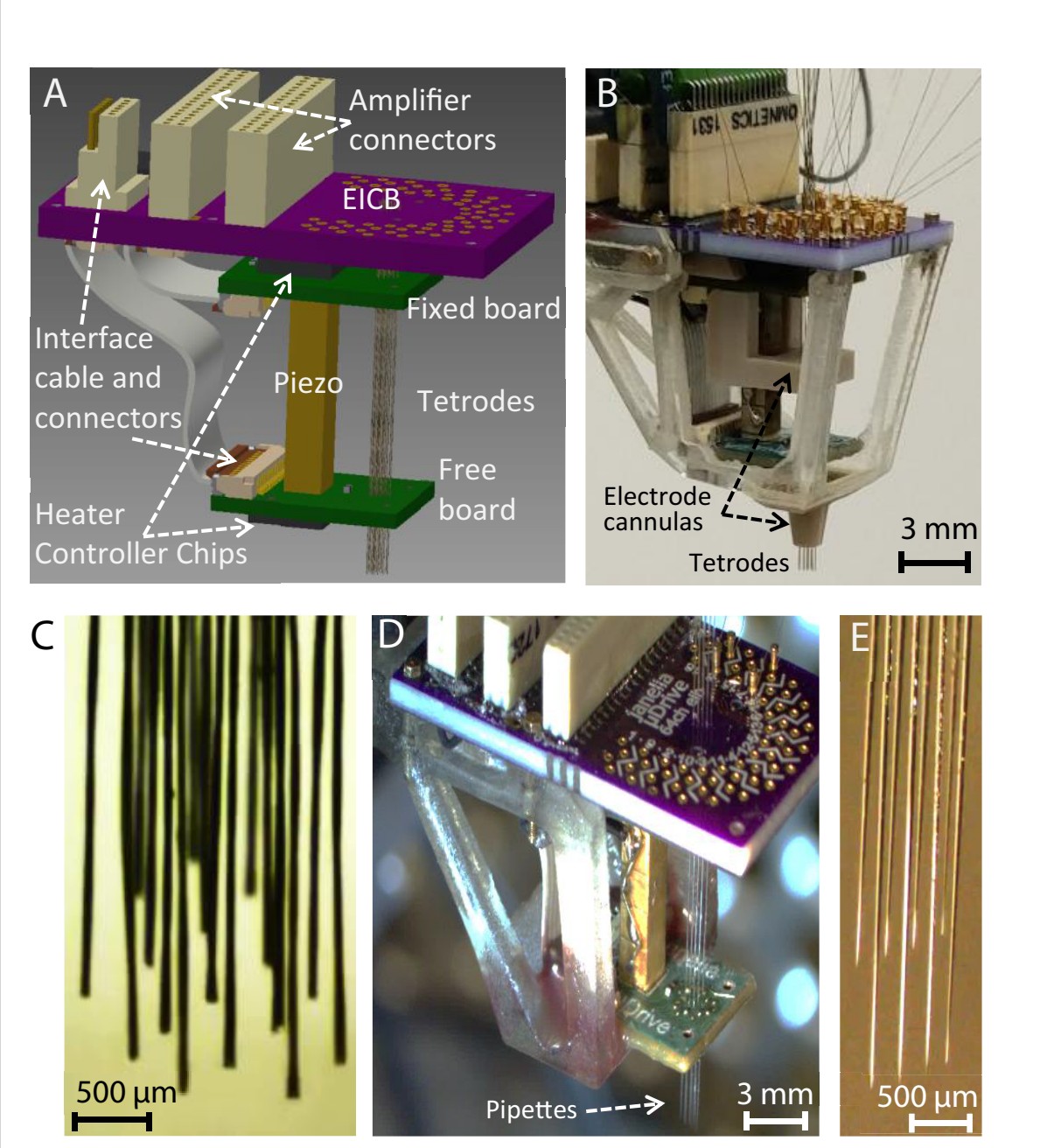

**Figure 5.** Implantable multi-probe single-actuator (MPSA) microdrives for neural recordings. (**A**) Schematic showing the main MPSA microdrive with the added electrode interface and control board (EICB). (**B**) Photograph of the implantable 16-tetrode-drive for acute neural recordings. Cannulas were added to increase tetrode stiffness during insertion. A three-dimensional-printed cage was added on the exterior for protection and ease of handling. The tetrode-drive was used to independently position 16 independent tetrodes shown in (**C**) using a single piezo actuator (*Video 4*). (**D**) Pipette-drive loaded with eight glass micropipettes, and independent motion of each micropipette (**E**) was also demonstrated (*Video 5*).

off-axis, i.e., laterally (*Figure 4F* 'cross-talk in air'). We reasoned that these momentary deflections of a stationary probe are likely caused by a combination of (1) thermal and phase change expansion of the PCM within the released bottom microgripper, (2) lateral motion within the bore while released, and (3) a non-straight and non-cylindrical probe (e.g. tetrode) sliding against the inner wall of this microgripper. We therefore hypothesized that much of this cross-talk could be reduced by mechanically dampening the pipette tip, as would occur in the brain. As expected, when the same test was performed in an agar brain phantom, the stationary pipette moved significantly less

(*Figure 4F* 'cross-talk in agar'; on-axis (x): 1.1±0.2 μm, p=0.03; off-axis (y): 0.85±0.2 μm, p<0.0001; n=5 steps; Student's unpaired t test). This experiment was also repeated with three tetrodes moving in agar where the maximum observed off-axis deflection of the stationary electrode across >100 steps was ~6 μm, and the on-axis deflection was negligible (*Video 3*). In a separate experiment, we continuously moved a tetrode and quantified the slow drift of the neighboring stationary tetrode. After the moving tetrode was stepped continuously for 90 s (total distance of 500 μm), the stationary one was found to have drifted less than 2 μm in x and y in air as well as in agar (*Figure 4G*). To our knowledge, no such equivalent experiments have been performed with manual microdrives.

Finally, the maximum weight that can be supported by the microdrive grip on a tetrode before the microgripper hold breaks was measured to be 58.0±2.4 g (n=5 tetrodes; *Figure 4H*), equivalent to a holding force of 0.57 N.

## Multi-probe single-actuator inchworm microdrive neural recordings

An integrated robotic microdrive for neural recordings with 16 twisted wire tetrode electrodes was assembled as shown schematically in *Figure 5A*, and in the photograph of *Figure 5B*. In addition to the main MPSA inchworm microdrive structure described previously, the complete design included an electrode interface and control board (EICB) with 64 recording channels (4 channels per each tetrode), connectors for extracellular recording amplifiers, electrode cannulas for maintaining electrode alignment, and a mechanical cage and mounts for encasing and positioning the drive for acute and chronic recordings. We again demonstrated the independent translation capability of the microdrive using only a single piezo actuator with 16 twisted wire tetrodes (*Figure 5C*, *Video 4*). The sequence of electronic actuation signals sent to the sets of 16 microgrippers in the top and bottom boards (as described in *Figure 2*) independently controlled the direction of motion of each twisted wire tetrode.

As mentioned earlier, the microgrippers of the device can grip probes of any material and cross-sectional shape, as the liquid PCM in the gripper bore conforms to the shape of the probe before solidifying. For the purposes of demonstrating this feature of the microdrive, in addition to neural twisted wire tetrode electrodes, we also independently translated eight glass micropipettes (50 μm inner diameter/80 μm outer diameter) while only modifying the size of the heater coil (~100 μm inner diameter/135 μm outer diameter) (*Figure 5D–E*, *Video 5*).

We first demonstrated the neural recording capability of our microdrive in acute surgical settings on anesthetized rats (see Materials and methods), initially with 4 twisted wire tetrodes (*Video 6*) and then with 16 twisted wire tetrodes as shown in *Figure 6A*. Under remote operation, the electrodes (*Figure 6B*) were independently advanced in 9.5 μm steps sequentially into the brain, while 64 channels (from 16 tetrodes) of neural signals were monitored visually in order to guide the approach of the tetrodes toward the targeted CA1 layer of the rat hippocampus. In total, each probe moved ~2.5 mm, or ~250 steps each for a total of ~4200 step sequences within the experiment. While the microdrive is capable of advancing the probes together, due to tissue friction and compression, it is preferable to progress each individually (or in small numbers). This is particularly simple to implement as the probe movement patterns can be adjusted and scripted through software, as opposed to manual handling of the implant on the rodent. If done with a manual microdrive, this would have induced significant mechanical artifacts into the neural signal used for position feedback, whereas our device produced only ~10 ms of interference within the spikeband after each step (*Appendix 1—figure 6*), and ~0.1 s overall, allowing for rapid and ergonomic optimization. Otherwise, there was no observed interference from the presence of the microdrive. In addition, these artifacts are far shorter than the duration an experimenter would typically wait to observe any effect (>5 s) of advancing an electrode, particularly when accounting for releasing the animal after manual adjustment. *Figure 6C* shows the sequential tuning of 16 tetrodes to the CA1 layer of the rat

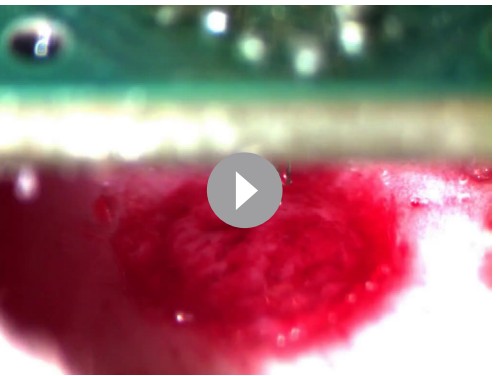

**Video 6.** Insertion of four independently controlled tetrodes into the rat brain through a craniotomy.
https://elifesciences.org/articles/71876/figures#video6

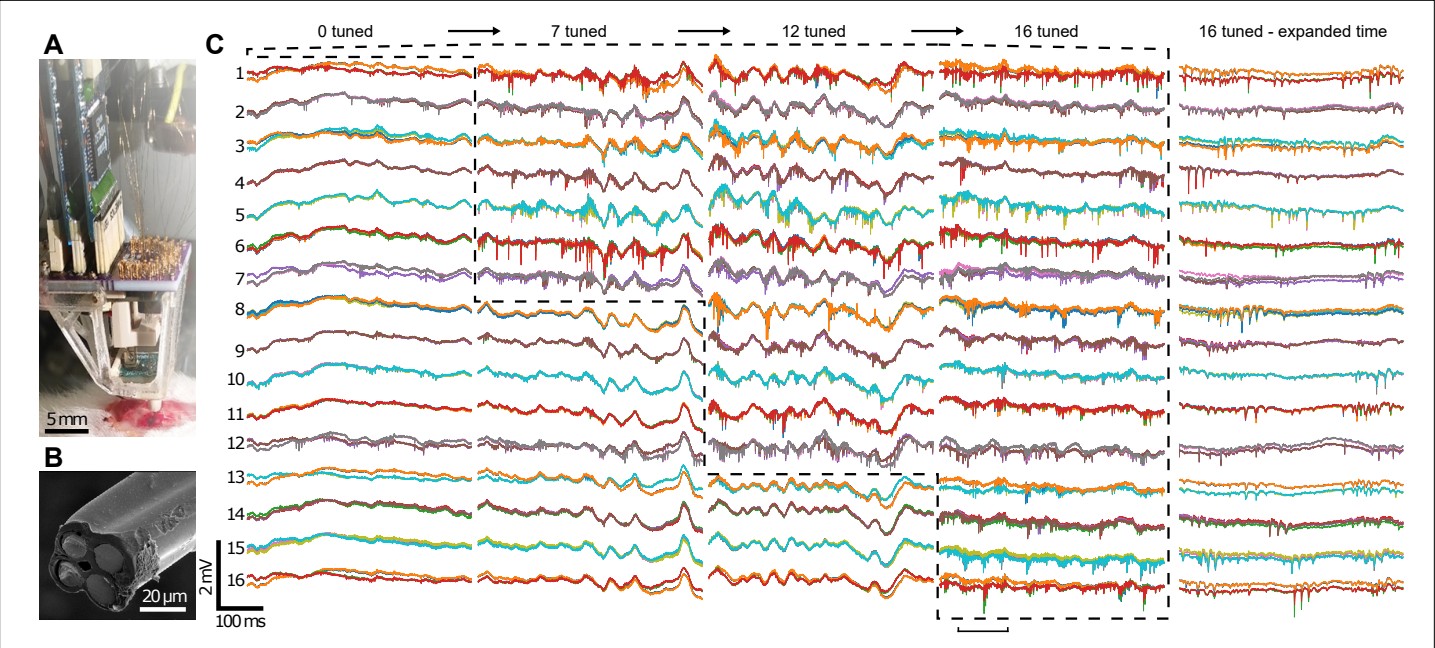

**Figure 6.** Neural recording with the implantable 16-tetrode-drive in an acute surgical setting. (**A**) Photograph of the acute experiment (craniotomy at bottom). (**B**) Scanning electron microscope image of the cut-face of a tetrode with the four exposed neural recording sites. The tetrodes were independently advanced in 9.5 µm steps sequentially into the brain. (**C**) Voltage traces from 64 recording channels (4 channels per tetrode) during precise depth tuning, beginning after all tetrodes, were brought near CA1. The position of each tetrode was independently adjusted to optimize its signal; dashed lines show groups of probes that have been optimally tuned. Previously tuned tetrodes did not lose signal after tuning of neighboring electrodes. For each of the 16 tetrodes, the 4 traces from each electrode (different color for each) of a tetrode are overlaid. Rightmost, time expanded segment of traces from period marked by horizontal bar at bottom. The total time to go from 0 to 16 tetrodes tuned here was ~1 hr 10 min.

hippocampus. In the first sub-panel, the 16 twisted wire tetrodes have already been moved to just above the CA1 layer, where they detect mainly the local field potential fluctuation in the brain and only small neural spikes. The sub-panels that follow show clearly distinguishable spiking activity after the tuning of 7, 12, and all 16 tetrodes. Importantly, tuning of additional tetrodes did not compromise the quality of the already placed tetrodes, as desired. In the fourth and fifth panels, the recordings after all 16 twisted wire tetrodes had been tuned are shown, with high-quality neural spike signals from all 64 channels.

As a final demonstration, we performed chronic neural recording in an awake, unrestrained rat using an implantable tetrode-drive with four loaded tetrodes (*Figure 7A*). Before application of dental acrylic to cement the device to the skull, the overall weight was 4.5 g with overall dimensions of 25×15×31 mm. As mentioned above, the weight and size would not have changed significantly if all 16 probes had been loaded: the overall structure would have remained the same, with the only additional weight coming from the 12 additional tetrodes and their connecting pins, grippers, and PCM, which we estimate to total approximately 10 mg (~50 mg if adding an optional intermediate spacer to reduce lateral bending of the flexible probes). The microdrive was maintained for 6 weeks, and the tetrode position was periodically adjusted through remote electronic control for fine-tuning of neural spike signal quality. The recordings exhibited expected sharp-wave ripple and spiking activity (*Figure 7C–D*). During chronic implantations, a back-and-forth progression was used to advance the tetrodes to ensure minimal tissue movement after final probe alignment, leading to at least ~700 steps per probe and ~2800 steps for the microdrive. Of particular importance was that, during these experiments, the animal was only handled at the beginning and end of the measurement session, when the electrode and control cables were connected, and then all probe adjustments were controlled remotely through a computer. We would like to emphasize that this process would normally require the experimenter to end the current behavior, restrain the awake animal sufficiently to grasp the typically quite small actuators, and then carefully manually adjust them while also avoiding damage to the typically exposed probes. In addition, such manual adjustments are not as precise, because of limitations

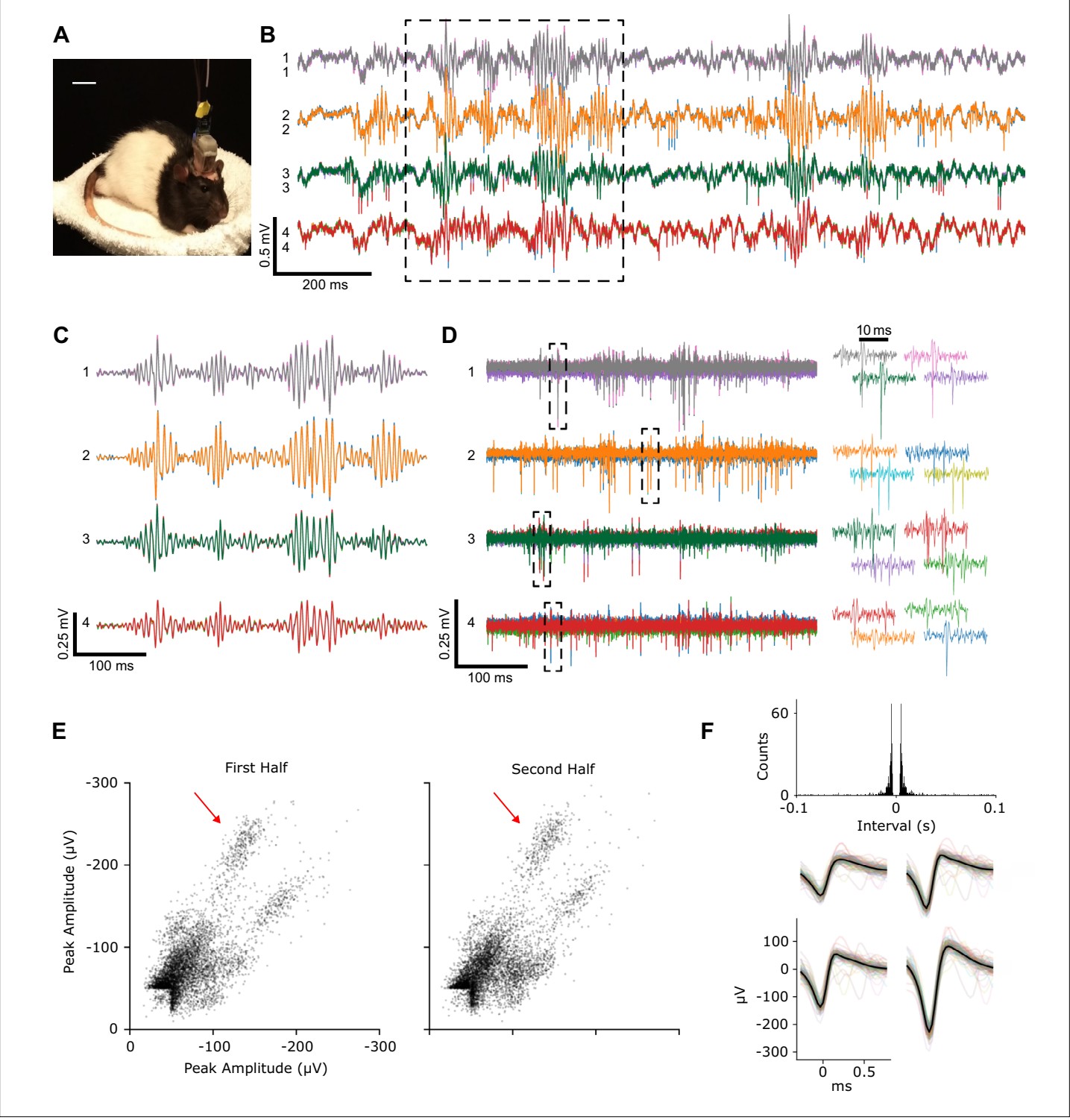

**Figure 7.** Recordings from chronically implanted head-mounted multi-probe single-actuator tetrode-drive during unrestrained, free behavior of an animal. (**A**) Rat implanted with the tetrode-drive. (**B**) Sample raw voltage traces from four tetrodes obtained ~6 weeks after implantation showing LFP, sharp wave ripples, and spiking activity in the hippocampus. (**C**) Boxed voltage segment from (**B**) filtered to isolate sharp-wave ripples (bandpass: 100–300 Hz). (**D**) Boxed voltage segment from (**B**) filtered to isolate action potentials (bandpass: 500–8000 Hz). For each of the four tetrodes, the four traces from each electrode (different color for each) of a tetrode are overlaid. Right: spike waveforms from boxes at left. (**E**) Peak amplitude of each spike recorded on two of the four channels of tetrode 1 during the first and second halves of an ~17 min period between successive tetrode adjustments. (**F**) Inter-spike interval histogram and waveforms of the spikes on the four channels from one unit (marked by arrow in E) during the full period in E. Average waveform on each channel (thick black line) and individual spike waveforms (thin colored lines) of 100 evenly spaced spikes of that unit.

to manual control and, in screw-based drives, because of backlash. Spike amplitudes and waveforms were stable during periods between successive tetrode adjustments, allowing isolation of multiple single units during free behavior in chronically implanted animals (*Figure 7E–F*, *Appendix 1—figure 7*, and Methods).

## Discussion

We have described the concept and use of a new robotic MPSA neural microdrive. The microdrive features a novel electronically controlled microgripper that utilizes temperature-controlled PCM as a gripping medium (*Barbic et al., 2017*). Using the gripper's microscopic dimensions, independent electronic actuation control, and high packing density, we demonstrated robotic placement of multiple independent neural recording electrodes into the CA1 region of the rat hippocampus as well as high-quality single-unit recordings from that area in the unrestrained, freely behaving animal. This was done with micrometer precision and unlimited probe translation capability, using only a single piezo actuator in acute and chronic in vivo settings. We emphasize that by taking advantage of the benefits of the MPSA concept, here we have detailed a first demonstration that this concept can provide the basic microdrive functionality of moving a practically significant number of probes to obtain high-quality signals with chronic implant durability, while also providing unique probe versatility, scalability, and remote control. It is expected that optimizations for applications not presented here will maintain these benefits of the core MPSA concept.

We highlight several advantages of our MPSA concept and implementation over traditional neural microdrives. First, the travel range of the MPSA microdrive is conceptually only limited by the length of the probe. Therefore, the MPSA microdrive could be used to simultaneously target distant regions of the brain (e.g. cortical and thalamic brain tissue) in large rodents or even primates (*Dotson et al., 2017*). Second, the MPSA concept makes the microdrive inherently more scalable than traditional microdrives. Each additional channel in a traditional motorized microdrive typically requires a motor, shuttle, and connector, causing the weight and size to increase substantially with channel count. On the other hand, in the MPSA microdrive, the addition of an extra channel causes a negligible weight and size increase, since the need for independent actuators is removed. Third, further improvements in scalability will rely on advances in the miniaturization of microchips and other electronic components (e.g. higher-channel count current source integrated circuits), which is likely to outpace the miniaturization of traditional motors. Fourth, traditional microdrives utilize an inverted cone geometry in order to accommodate bulky actuators. This arrangement accommodates the use of flexible neural probes such as twisted-wire tetrodes and long silicon shanks (*Michon et al., 2016*) but precludes the use of stiff micropipettes and short silicon shanks which are typically too inflexible to bend along the angular guide channel and through the vertical cannula. Since the microgrippers in the MPSA microdrive can be tightly packed, the conical geometry is not needed, and rigid probes such as micropipettes can be translated.

Another fundamental feature of the MPSA microdrive is that the releasing and gripping of the electrodes during stepping are gentle. The method is based on the melting and solidifying of the PCM around the electrode and inside the PCB via no matter what the shape or material of the electrode is (e.g. tetrode, glass capillary, silicon probe, and optical fiber). This mechanism of gripping and releasing of electrodes ensures that there is no large or sudden gripper movement (as would perhaps be expected from other mechanically based grippers) that might potentially damage the electrodes. During the course of our work, we have never observed any mechanical or electrical damage to the recording electrodes that in any way compromised our electrical neural recordings. It is also important to emphasize that during the motion of the electrodes in our device, as diagrammatically shown in *Figure 2*, the MPSA mechanism does not cause additional axial strain or compression on the electrodes, which would perhaps cause damage if they were. During the electrode motion while stepping, the electrode is only held by one and only one of the electrode grippers (either top or bottom) but never both. Therefore, extension or contraction of our piezo actuator itself during motion never extends or contracts the electrode itself. Furthermore, the electrodes held and positioned by our device are never fully loose while being translated. They are always gripped by at least one gripper at any one time during translation, as *Figure 2* describes. This ensures the stability of any one electrode while the adjacent electrodes might be positioned as needed.

Minimal thermal design is required for a single probe implementation but coupling and heat flow become more important as the probe count and heater density increase. The relevant phenomena, however, can be readily modeled with heat transfer simulations, and we envision that more sophisticated mechanical and control schemes could yield higher probe densities and movement speeds (see Appendix 1 for additional details). In particular, increasing the movement speed is important for generalizing the use of these temperature-sensitive PCM-based grippers, as many applications require faster rates than utilized here. Examples of more advanced methods include integrating copper heat sink regions between microgrippers and optimizing heating timings based on the mutual impulse response and heating history of all the thermally coupled heaters on a board. Additionally, the PCM could be chosen for the temperature of the application or designed to have properties helpful to the microgripper functionality such as higher thermal diffusivity and a narrower phase transition (*Hyun et al., 2014*).

It is important to emphasize that the heating of the brain due to operation of our device is essentially negligible. During the robotic stepping of the device for electrode translation, described in *Figure 2*, the heating pulse required for melting of the phase change grippers is extremely short (~10 ms), while the piezo actuator's steps (which do not generate heat) are ~1 s long. Therefore, the heaters in any of the PCBs of our device are on for only up to 1% of the overall stepping motion time. Additionally, that small amount of heat energy generated by the heaters within the PCB vias is (by design of the copper layers of the PCB) rapidly dissipated within the large PCB volume and does not thermally conduct to the brain through the very large thermal resistance of the microscopic recording electrodes. Furthermore, the PCM for this device was chosen so that the grippers would not need to rise to a temperature significantly above the typical temperature of the brain. Finally, during the electrode recordings from neurons, or simply during the idle state of the device, all the electrodes are fixed (gripped) in the device, which by design occurs when all the heaters in the device are off and generate no heat.

Electrical pick-up artifacts during electrode motion are also of potential concern, as is to be expected with nearby applied voltages, electrical currents through the microheaters in the PCB vias that surround the recording electrodes, and repeated piezo actuation steps (*Appendix 1—figure 6*). However, the electrical recordings from neurons with sensitive neural amplifiers coupled to the electrodes were regularly performed during the stepping motion of the device without apparent issue to the tissue or electronics. It was never necessary to disconnect or power down the elements of the device that perform the motion steps (piezo actuator and the heaters) to improve the data quality, but if motion was not being performed during a recording session, it was also not necessary to use or power these elements. Therefore, the described MPSA mechanism in no way affects the ability to perform effective neural recordings. As recordings can be taken with high fidelity after each motion step of only ~10 microns or less, this mechanism is instead likely to yield improved recording quality.

While this MPSA microdrive design is particularly attractive for rodent electrophysiology where miniaturization is critical, in other uses, weight and size can be traded for additional functionality. For example, in situations where motion accuracy is critical, a positional feedback system such as a small camera could be installed. Alternatively, the design does not require ferrous components, and therefore an MRI compatible version of the microdrive (*Matsui et al., 2007*) would be a particularly interesting variant of imaging feedback control. Further directions for the implantable neural MPSA microdrive include enabling untethered control and recordings through wireless communication via Bluetooth or Wi-Fi, implementing on-board piezoelectric drivers and battery-powered operation.

An important advancement to the functionality would be the implementation of autonomous probe positioning. While in our demonstrations the operator was involved in the probe positioning, computer-controlled closed loop spike detection and fidelity monitoring capabilities would allow for the placement of a probe within a region of interest based upon a model signal and thereafter optimization of signal quality, in which degradation could trigger automatic compensatory motion (*Cham et al., 2005*). Such automation could lead to both improved data yield and reduced operator time in experiments.

Apart from neural electrophysiology probes, the MPSA microdrive could also be used for other biological as well as non-biological probe translation tasks (*Matsui et al., 2007*; *Sych et al., 2019*). For example, the microgrippers could be sized to accommodate hypodermic needles, optical fibers, and ultrasonic transducers and sensors on a variety of scales. For any of these larger diameter probes,

minimizing the amount of PCM within the via between the probe and the helical heater coil would be important. This would reduce the required heat to melt the PCM, reduce the lateral motion of the probe during translation, and maintain the capillary action that keeps the PCB within the via. The heater coil would be correspondingly carefully sized to satisfy those requirements. With larger probes, there becomes an additional consideration of the amount of heat the probe itself can conduct away from the gripper. This is based on the probe's cross-section and thermal diffusivity characteristics, where a very thermally diffusive probe might require modifications to the process (such as higher power and a relatively longer melting time) in order to fully melt the PCM in the via. Another possibility would be to slightly modify the probe by adding a less thermally diffusive sheath around the probe that is instead in contact with and delays heat loss from the PCM. The key idea would remain the use of a short heat impulse so that the heat minimally diffuses into the probe relative to the PCM during translation.

In conclusion, our robotic microdrive provides the capability for independent positioning of multiple parallel probes using only a single piezo actuator with significant benefits wherever the need arises for small weight, miniature device size, sub-micron level probe positioning precision, and remote control.

## Materials and methods
### Microgripper construction
Heating coils were prepared by winding fine insulated nichrome wire (California Fine Wire, Model Stableohm 800 A, 12.5 µm diameter conductor, 17.5 µm diameter with insulation) around a 75 µm diameter tungsten wire core (California Fine Wire). The final microheater had ~45 turns, as shown in *Figure 1C*, and had a typical resistance of 125 Ω for a tetrode microgripper. Docosane (Sigma-Aldrich, item 134457) was chosen as the PCM due to its non-toxicity and sharp, low melting temperature (~42°C), which is still above animal body temperature (*Balaban et al., 2005*).

### Thermal characterization
The gripper board designs were simulated in COMSOL Multiphysics before fabrication in order to implement a design with sufficient microgripper thermal independence (see Appendix 1). Independence was assessed based on whether the temperature in the 'off' microgrippers remained below the lowest temperature of the phase transition region (~41°C) when the 'on' microgrippers could reach a temperature above the highest temperature of the phase transition region (~44.5°C). An additional safety margin of ~5°C for both temperatures was included when analyzing the simulations to conservatively account for potential differences between the model and the fabricated device. Simulations were performed with an initial temperature of 20°C, and a smaller safety margin would be used for initial temperatures closer to the phase transition temperature to reflect the smaller overall changes. Heat duration profiles as a function of board temperature were also obtained due to expected temperature changes in the environment and during use. Timing parameters were ultimately adjusted based upon empirical movement tests with assembled microdrives of varying probe counts.

### MPSA microdrive motor assembly
Gripper boards were manufactured using standard PCB technology (8×11×0.8 mm, six layers, Sierra Circuits). Each board contained an integrated current source chip (STMicroelectronics LED1642GW), associated passives, 0201 thermistor, and a flat flex cable connector. Each current source chip was pre-characterized in a custom, automated setup in order to use only chips with consistent and high current across all channels. These components were soldered onto the board before drilling of the microgripper vias. To increase the microgripper and overall board cooling rate (and therefore the cycle rate), solid internal layers and thermal vias (~27 per cm$^2$) in the non-gripper regions of the board were used to increase the thermal conduction across the plane and through the thickness of the board, respectively (see Appendix 1). Additional through-holes were placed on the board and used for alignment and mounting purposes when integrating these boards into larger structures. Via holes (4.7 mil for tetrodes, 5.9 mil for pipettes) were drilled into the PCB in a hexagonal close packed array with 300 µm spacing using a five-axis CNC mill (Hermle). The microheaters were inserted into the via, glued in place with high-temperature epoxy (Thorlabs, 353NDPK), and the core tungsten wire was extracted for a final open bore resistive via, shown in *Figure 1D*. The heating coils in each board were

individually soldered to exposed pads, which were routed to the independent channels of the current source chip. The connection pads and leads were insulated with epoxy in order to yield a device, which was mechanically robust to handling and contact near the heaters, as would occur during the loading and cleaning procedures. To complete the microdrive assembly, the piezoelectric stack (Thorlabs PK3CMP2) was positioned on the designated locations on each board and epoxied to each, one at a time. A custom jig with alignment pins (McMaster-Carr) was used during the curing process in order to ensure that the boards were aligned and parallel.

Building the core MPSA motor – the grippers, piezo, and board assembly – and developing the control software are in principle possible for a typical system neuroscience-focused laboratory. The materials are available, the engineering techniques are not prohibitively difficult, but the process is specialized. In our practice in a research and development laboratory setting, a construction of the entire device takes about 1 week of labor for a trained person. However, once the MPSA motor is constructed, it is reusable, and the preparation time of the microdrive for experimental use is only required for reloading of the recording electrodes.

## Implantable microdrive system

In addition to the MPSA motor described above, the implantable microdrives (tetrode-drive and pipette-drive) necessitated an EICB, a base station, and a PC application for control.

The EICB provided crimping locations for 64 electrodes and 2 Omnetics connectors for 32 channel Intan amplifier boards (Intan Technologies, C3314, *Figure 5B*). These were ordered in 16 sets of 4, for a maximum of 16 tetrodes. There were additional contact locations for reference electrodes and grounding connections. In the center of the connection region, a pair of alignment holes were included to match the ones used on the gripper board. This facilitated alignment of the boards and the drilling of probe vias in the EICB.

The control section of the board consisted of a microcontroller (Microchip PIC16F18345), three-color LED indicator (Rohm Semiconductor SMLP36RGB2W3R), and connectors for the base station cable as well as the two flex cables going to the heater boards and connection pads for the piezo wires. The microcontroller implemented the timing and coil heating operations determined by the movement set in the PC software. Thermistors were used to measure the temperature of each board to account for onboard heat dissipation and compensate for animal body heat. The temperature was fed back to an algorithm in the PC software and used to adjust heat durations. The cable to the base station consisted of 5 m or shorter lengths of thin, unbundled wires (Cooner Wire, Stranded Bare Copper FEP Hookup Wire). For connecting the gripper boards to the EICB, custom length flat flex cables were made by shortening a siliconeflex cable to the proper length and adhering plastic backing to allow insertion into the connectors.

The base station consisted of a USB-connected microcontroller (Arduino Mega 2560), piezo driver (Thorlabs KPZ101), heater power circuit, and connectors. A trigger line from the EICB was monitored by the base station microcontroller in order to command the piezo driver with highly accurate timing.

The PC control application was implemented in LabVIEW (National Instruments). It provided the ability to select the probes to move and the direction to move each probe. The application had the capability for scripted series of movements, changing the frequency of movements, the current through the heating coils, durations of heating, and temperature compensation. It also produced a record of the movements and displayed the total movements and offset from a set point.

## Implantable microdrive assembly

For the tetrode-drive, a custom-milled, temperature stable polyether ether ketone (PEEK) spacer was used to join the top gripper board and EICB to a three-dimensional (3D) printed superstructure cage, which protected the microdrive from damage and allowed for easier handling. Although the high aspect ratio of the bores maintains good probe axial alignment in the microgrippers, we realized that flexible nichrome wire tetrodes bend easily, which can reduce or negate the length of each step. We therefore included two guiding PEEK cannulas. The mid-drive and bottom-tip cannulas were aligned with the top and bottom microgripper vias using guide pins and attached to the top board and cage, respectively. The bottom board was chosen to be mobile in order to minimize bending by minimizing the length of tetrode between the piezo-coupled microgripper and the brain. For chronic testing, the EICB was also enclosed with 3D printed caps, and the open sides of the cage were sealed with plastic

film sheets. The mid-drive cannula was not used in the chronic experiment. M0.6 fasteners (Prime-Miniatures) were used throughout for modifiable attachments.

During acute experiments, it was seen that evaporation from the craniotomy condensed on the bottom board. For this reason, the bottom board was insulated with a conformal silicone PCB coating (MG Chemicals 422B) throughout the non-heating area.

## Probe loading

Twisted wire tetrode recording electrodes were prepared from fine insulated wires (California Fine Wire, Model Stableohm 800 A, 12.5 µm diameter conductor, 17.5 µm diameter with insulation). Each electrode was loaded into the microdrive by threading it through the empty microheater bores and cannulas with fine tweezers under an inspection microscope. A small particle of solid PCM (sized to underfill the probe filled bore) was placed near the microgripper bore either on the board or on the threaded probe with fine tweezers. The PCM was melted either using the microgripper's heater or a heat lamp so that it could flow into and fill the bore through capillary action. One end of the tetrode was cut with a sharp razor blade in order to expose the four neural recording sites (shown in *Figure 6B*), while on the other end each wire was crimped separately into the EICB for connection to the digital recording system (*Figure 5B*). When loading many probes at high density, the probes were left loose in their bores until all were threaded, after which the PCM on the probes was melted and slid into the bores. Thereafter, the PCM in the microgrippers was melted during any additional probe handling (e.g. trimming and site exposure). Previously, leaving the probes mobile in the bores avoided damage to the probes due to contact with the tweezers and also better controlled the amount of PCM in each microgripper by preventing flow into neighboring bores. Because the EICB did not need to be enclosed during acute experiments, the tetrodes were left long between the crimp locations and the EICB probe vias. Sixteen tetrodes could be loaded over 2–3 days. Any laboratory that is capable of loading conventional screw-based tetrode neural microdrives would be capable of loading probes into an MPSA microdrive. Loading of the robotic device with tetrodes is similar in technical complexity, effort, and time to loading of the classic manual tetrode drives.

The same procedure was used to load the microdrive with the glass microcapillaries (Vitrocom Model CV0508 50 µm ID/80 µm OD borosilicate glass) for the glass capillary microdrive translation demonstration. Eight pipettes could be fabricated and loaded within half of a working day.

## Mechanical testing setup

To characterize the motion of the microdrive, it was first rigidly mounted in a 3D-printed clamp such that probes moved horizontally. Probes (pipettes or tetrodes) were loaded into the drive as described above. The maximum piezo displacement voltage (100 V) was used for all tests unless otherwise noted. For accuracy characterization, the MPSA microdrive loaded with a single probe was placed under a USB microscope (VMS-004, Veho), such that the probe was in the field of view. In all other motion characterization experiments (repeatability, resolution, step size, cross-talk), the assembly was placed under a probe station microscope (Alessi REL-4100A) equipped with 10× and 60× objectives (Mitutoyo) and a digital camera (EO USB 2.0, Edmund Optics). In the cross-talk experiments, the microdrive was loaded with two probes. In a subset of experiments, probes moved through 0.2% w/v agarose (Millipore Sigma). Bidirectional repeatability was measured by repeatedly commanding a probe to move to a single target position in the travel range. The target position was approached from 0.4 to 5 mm away from both directions.

For the microgripper holding force characterization, we used a load cell (RB-Phi-203, RobotShop), which we first calibrated with known weights. One end of the load cell was attached perpendicular to a linear translation stage (PT1, ThorLabs). A 3D-printed bracket with a metal bolt was attached to the other end. The free end of a tetrode in a stationary microdrive was superglued to the bolt. The translation stage was used to pull the tetrode until it came loose from the microgripper. The maximum force on the load cell immediately before the break was considered to be the holding force of the microdrive. All image analysis was performed using FIJI (*Schindelin et al., 2012*).

## In vivo electrophysiology

All procedures involving animals were performed according to methods approved by the Janelia Institutional Animal Care and Use Committee. All acute surgical experiments were performed on

anesthetized Wistar male rats (age: P20–P30). The animals were placed in a stereotaxic apparatus maintained at 37°C with the anesthesia maintained at 1.5–2% isoflurane. A small craniotomy (diameter ~1 mm) was drilled in the skull (position coordinates for dorsal hippocampus: 3.5 mm posterior of bregma, 2.5 mm lateral of midline) and the dura removed with a sharp needle in a small region (diameter ~0.25 mm) centered in the craniotomy. A chlorided silver wire was folded under a flap of skin near the incision and used as a reference electrode for the recordings. The microdrive loaded with 4 or 16 tetrodes was positioned over the center of the craniotomy using a motorized micromanipulator (Luigs & Neumann) and a 3D printed holder.

To record from the CA1 pyramidal cell layer, probes were advanced approximately 2.5 mm from the surface of the cortex. The cells in this layer produce pronounced and distinct burst patterns and remain relatively active under anesthesia. Furthermore, electrodes progressing toward the cell layer pass through regions with characteristic activity patterns. Using these indicators, the probes were moved to and stopped at the cell layer. When near actively firing cells, the effect of probe adjustment was typically seen immediately, though there is a slow tissue relaxation component that was additionally accounted for. Smaller step sizes near the cell layer were achieved by adjusting the step voltage on the piezo actuator.

For chronic recordings, Long-Evans rats were first anesthetized with isoflurane and headfixed in a stereotaxic frame. A craniotomy was made over the CA1 field of the right dorsal hippocampus (AP –3.8 mm, ML 2.4 mm), and the dura was removed. The tetrode tips were previously gold-plated to reduce the impedance of each channel to <250 kΩ at 1 kHz, and a stainless steel screw placed in contact with the left cerebellum served as the reference electrode for recordings. Before implantation, the tetrode tips were retracted 50 µm into the bottom cannula of the microdrive which was then filled with melted Vaseline to prevent clogging by blood or CSF. In order to prevent Vaseline from contacting the microgrippers, the retraction of the probes was limited to 2.5 mm from their deepest point. The bottom cannula was then aligned to the craniotomy using a stereotaxic holder which connected to one of the Omnetics amplifier connectors of the EICB, and the microdrive cage was fixed on the skull with OptiBond (Kerr), A1 Charisma (Kulzer), and dental cement (Lang). Soon after implantation, the depth of sharp wave ripples was found for each probe, which was then retracted and re-descended over several weeks until sharp-wave ripples and spikes were clearly seen (*Rich et al., 2014*). As with manual positioners, re-adjustment is expected to be occasionally required. At the termination point of a chronic experiment, the EICB and microdrive motor were recovered for future use.

Neural recording data from 64 total channels from 16 tetrodes was digitally sampled at rates of 20 kHz/channel for acute experiments and streamed to the computer via USB using an Intan USB Interface Board. For chronic experiments, 16 total channels from 4 tetrodes were digitally sampled at 30 kHz/channel and streamed to the computer via USB using an OpenEphys Acquisition Board.

For isolating single units from the chronically implanted animal, we processed an ~17 min period between successive tetrode adjustments by the MPSA microdrive (*Figure 7E–F* and *Appendix 1— figure 7*). We first bandpass filtered the raw continuous recordings from each channel between 600 and 6000 Hz, then subtracted from each channel the average trace from the channels of the other tetrodes, and then extracted the spikes. The spike waveforms (32 samples at 30 kHz sampling rate, peak at sample 8) were extracted when at least one of the four channels of the filtered trace crossed the threshold of –50 µV. Only the largest amplitude spike was taken if multiple threshold crossings occurred within a 1 ms window. We took the peak amplitudes of the waveforms on each channel for each spike as the features and then manually sorted the spikes into individual single units using MatClust (*Karlsson, 2022*). For each unit we checked the following spike sorting quality metrics: inter-spike interval histogram, isolation distance, and L-ratio (*Schmitzer-Torbert et al., 2005*). We isolated 11 single units from the 4 tetrodes. Note that we did not perform tetrode adjustment in a manner to optimize the number simultaneously isolatable single units; rather, we stepped through the CA1 pyramidal cell layer while observing the spike amplitudes in the continuous traces, and then used a period with a long gap between successive adjustments (the ~17 min period described here) to examine recording stability and unit isolation. However, based on the stability and quality of these clusters/units as well as our experience with unit isolation when using standard, manually adjustable screw-based tetrode drives, we are confident that the MPSA microdrive will yield as many clusters/units of as good stability and quality as standard drives.

## Acknowledgements

This work was supported by the Howard Hughes Medical Institute. We would like thank Bill Biddle for excellent scientific machining of the robotic microdrive parts, Brian Barbarits for help with extracellular electrode recording electronics, Steve Sawtelle for help with electronics and PCB design, Roger Rogers for help with 3D printing, and Jeff Magee and David Hunt for helpful discussions.

## Additional information

### Competing interests

Richard D Smith, Mladen Barbic: M.B. and R.D.S. have applied for intellectual property patent protection on the devices and principles described in this work ("Positioning apparatus and gripping apparatus", US20200046317A1). The other authors declare that no competing interests exist.

### Funding

| Funder | Grant reference number | Author |
| --- | --- | --- |
| Howard Hughes Medical Institute | | Albert K Lee |

The funders had no role in study design, data collection and interpretation, or the decision to submit the work for publication.

### Author contributions

Richard D Smith, Performed the thermal design and testing of the resistive heating coil micro-grippers and PCM, performed the circuit, microcontroller and PCB design of the gripper boards, EICBs and base station, designed the mechanical and system structure of the microdrive, and developed the software control for remotely operating the robotic microdrive. Connected the micro-grippers, and assembled all the microdrives. Loaded the microdrives with neural recording twisted wire tetrodes and glass micro-capillaries. Performed the acute surgeries and neural recordings experiments. Performed the chronic surgeries and neural recordings experiments. Performed data analysis and data visualization. Expanded and edited the manuscript with input from all authors; Ilya Kolb, Performed the micro-gripper and microdrive mechanical and probe positioning, coupling, and gripping characterization. Performed data analysis and data visualization. Expanded and edited the manuscript with input from all authors; Shinsuke Tanaka, Prepared the twisted wire tetrodes neural electrodes. Performed the chronic surgeries and neural recordings experiments. Performed data analysis and data visualization; Albert K Lee, Supervised the project and advised on the surgical procedures and neural recordings. Performed data analysis and data visualization. Expanded and edited the manuscript with input from all authors; Timothy D Harris, Supervised the project and advised on the surgical procedures and neural recordings; Mladen Barbic, Conceived the principal ideas of the phase-change-material heat activated via micro-gripper and multi-probe-single-actuator inchworm microdrive. Fabricated and installed helical micro-gripper heaters. Performed the acute surgeries and neural recordings experiments. Wrote the initial draft of the manuscript. Expanded and edited the manuscript with input from all authors

### Author ORCIDs

Richard D Smith ⓘ http://orcid.org/0000-0001-9384-105X
Ilya Kolb ⓘ http://orcid.org/0000-0001-5100-849X
Albert K Lee ⓘ http://orcid.org/0000-0003-4332-8332
Timothy D Harris ⓘ http://orcid.org/0000-0002-6289-4439
Mladen Barbic ⓘ http://orcid.org/0000-0003-1511-1910

### Ethics

All procedures involving animals were performed according to methods approved by the Janelia Institutional Animal Care and Use Committee (Protocol# 14-116, 15-122 and 17-158).

Decision letter and Author response
Decision letter https://doi.org/10.7554/eLife.71876.sa1
Author response https://doi.org/10.7554/eLife.71876.sa2

## Additional files

### Supplementary files
• Transparent reporting form

### Data availability
Raw data and processing scripts for the acute and chronic experiments depicted in Figures 6 and 7 have been deposited in Dryad (https://doi.org/org/10.5061/dryad.98sf7m0jj and https://doi.org/doi.org/10.5061/dryad.j9kd51cct).

The following datasets were generated:

| Author(s) | Year | Dataset title | Dataset URL | Database and Identifier |
|---|---|---|---|---|
| Smith RD, Kolb I, Tanaka S, Lee A, Harris T, Barbic M | 2021 | MPSA Microdrive Tuned Recordings from Acutely Implanted Tetrodes | https://doi.org/10.5061/dryad.98sf7m0jj | Dryad Digital Repository, 10.5061/dryad.98sf7m0jj |
| Smith RD, Kolb I, Tanaka S, Lee A, Harris T, Barbic M | 2021 | MPSA Microdrive Tuned Recordings from Chronically Implanted Tetrodes | https://doi.org/10.5061/dryad.j9kd51cct | Dryad Digital Repository, 10.5061/dryad.j9kd51cct |

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

# Appendix 1

## Gripper board inner layers and coil via gap

The gripper board contained four solid inner layers of copper for distributing the heat after each pulse with a gap around the via pattern of 0.25 mm (*Appendix 1—figure 1*). Due to the thermal diffusion time through the printed circuit board (PCB) laminate, the added copper at this distance would not practically affect the dynamics of the millisecond scale heating process but would contribute to cooling the gripper region over the 100 millisecond and greater time scale applicable to continuous stepping.

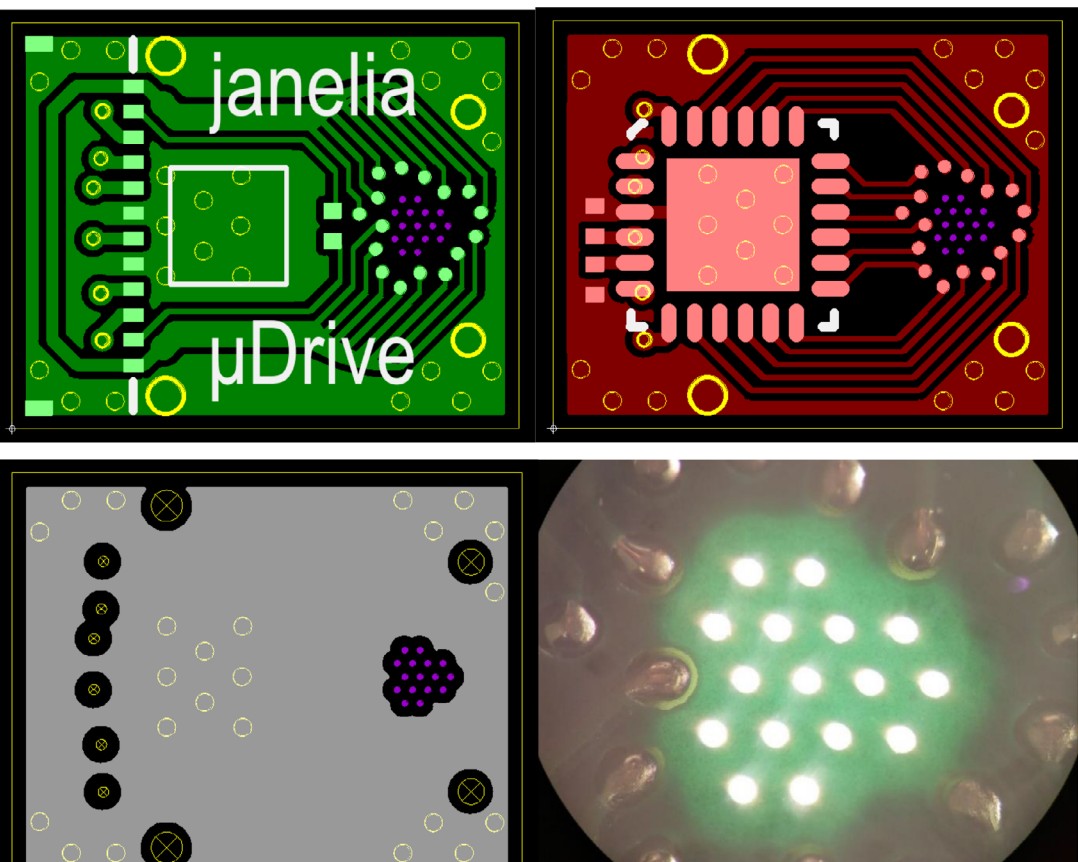

**Appendix 1—figure 1.** CAD images for the (top left) top, (top right) bottom, and (bottom left) inner layers of the gripper board, as well as (bottom right) an image of light passing through drilled vias and the non-copper areas of a fabricated gripper board.

## Coil installation details

Before insertion of the coils, trenches for the leads between the vias and their connection pads were laser-cut into the top surface of the PCB soldermask (*Appendix 1—figure 2*). After removal of the tungsten pin, the wire leads were placed into the trenches and covered with epoxy. This prevented damage to the fragile coil leads and outflow of phase change material (PCM) along the lead during heating. The insulation of the leads at the pads was then removed using the laser cutter. Since the coil wire metal does not wet solder well, the bare section of the leads was instead forced into a melted bead of solder on the pad, which would solidify and yield a compact electrical connection of negligible resistance. A controlled amount of solder on each pad was obtained from 0.25 mm diameter solder balls (Chip Quik Inc), and a narrow tip soldering iron was used to prevent bridging between adjacent pads. After soldering, the boards were sonicated in isopropyl alcohol to remove flux residue.

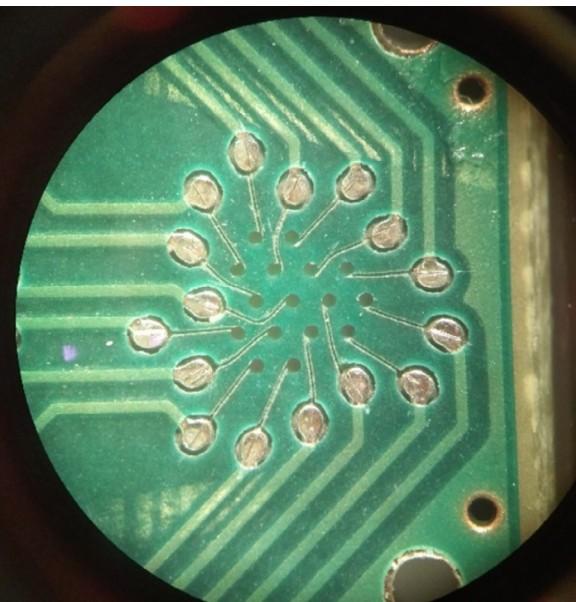

**Appendix 1—figure 2.** Image depicting the trenches cut into the printed circuit board solder mask for the heater coil leads.

## Example timing of a step sequence

For an implanted microdrive where the lower board nearer the animal is typically warmer than the upper board, the heating period for the lower board would be shorter, but both boards would end heating at the same time. For example, based on the board temperatures measured at their thermistor, the lower board may require 7 ms of heating and the top board 10 ms. In order to end at the same time, the lower board would begin heating 3 ms after the top board. Due to the propagation delay of heat through the PCM, a 2 ms delay could be applied after the ending of heating before the piezo was actuated. For a 1 Hz step rate, the heating for the second half of the cycle would begin 0.5 s after the first (*Figure 2*).

## Detailed description of tetrode loading for implantable devices

For each probe, approximately 15 cm lengths of twisted wire tetrodes were prepared beforehand, with one end snipped and the other end with the wires still separated. Initially, the cage, microdrive, and EICB were slightly separated and angled relative to each other such that the holes of the bottom cannula and top board were visible for threading. Under a microscope, the tetrodes were individually threaded through each via and stuck onto a piece of adhesive below the assembly to prevent it from withdrawing through the cannula. Throughout this process, tetrodes were only gripped by a section near the trimmed end that would later be trimmed away. Before threading another tetrode, two small particles of PCM were attached onto the tetrode near the boards, and then a heat lamp was used to melt them into beads. This process was repeated with all subsequent tetrodes. After all the tetrodes had been threaded, the cage, microdrive, and EICB were aligned and secured.

Each tetrode wire was then individually connected to the EICB. This can be done in the standard crimp style, but we instead used a slightly modified method to reduce the risk of incompletely shearing the insulation or completely cutting through the wire. The insulation was burnt off the connection ends of the tetrodes using a small heating element, and then the ends were wrapped and tied around the crimp pins and adhered using conductive silver paint. The pins were then inserted into their crimp holes, and stress was decoupled from the connection by applying wax over the pins.

The heating lamp was then used to heat the assembly and remelt the PCM on all the tetrodes. Each tetrode was pulled through the bottom cannula until only enough remained between the EICB vias and the crimp locations to allow for the desired range of motion. During this sliding process, the PCM beads were wicked into their bores such that they would grip the probe upon cooling. The ends of the tetrodes were snipped while the PCM was still melted in order to preventing any kinking during the cutting process.

The top cap of the EICB was then placed to protect the tetrode loops. Functionality of the tetrodes was checked by using the microdrive. If loaded successfully, there should be no kinks on the tetrodes, which would lead to bowing and poor movement behavior.

## Pipette-drive structure difference

As stated in the main text, the choice of moving and stationary gripper boards is dictated by the application. For the pipette-drive, we wanted to ensure that pipette tips would remain as stable as possible for potential juxtacellular or intracellular electrophysiology experiments. Therefore, in that design, the top board was mobile, while the bottom gripper board was fixed to a three-dimensional printed superstructure which also held the electrode interface and control board (EICB). In this configuration, the nearest board to the tips never releases a non-translating probe, whereas in the tetrode drive, the gripper in the lower board would heat and slide along the probe. No cannulas were added since the pipettes did not bow in practice.

## Potential failure modes

We note two potential failure modes which become more likely with longer movement lengths: (1) PCM may leak out from the microgrippers during use and (2) any liquid or tissue drawn up from the brain into a microgripper could render it inoperable. Both of these failure modes would be rare and preventable.

Leaking can be prevented by ensuring that the bore was not overfilled, and a sufficient amount of heat was applied to melt all of the PCM during the release. It is important to note that a portion of PCM leaking out of the bore does not prevent the gripper from continuing to function. Additionally, if the PCM that leaked out onto the probe was still near the board, it could be withdrawn back into the bore. This would be done by turning on all the heaters of one board at a low intensity such that the bores and probes would rise above the melting temperature, which would cause the PCM to flow back into the bore. Since the heaters of the other board could remain off, the probes would remain in-place such that this could be done post implantation. The heating and cooling times of this process would be substantial (~10 s) but could be integrated into an overall movement scheme if PCM loss mitigation was critical.

Prevention of foreign substances from reaching the heaters was accomplished by maintaining a maximum total retraction distance from the maximum depth at any point. During chronic implantation, this was done to prevent Vaseline from contacting the PCM, which it could dissolve or at least broaden the phase change temperature profile. Analogous limits could be set when doing acute implantations with or without the tip cannula.

## Gripper board simulation

The gripper board designs were simulated before fabrication in order to implement a design with sufficient microgripper thermal independence. The simulations were performed in COMSOL Multiphysics using the Heat Transfer module with the phase change material function. Because the region of interest minimally varies through the thickness of the board, two-dimensional simulations were used and set at the plane of an inner layer (*Appendix 1—figure 3*). The coils were modeled as ring-shaped heat sources, and their heat generation properties were applied as a volumetric average from their measured resistance and the currents passed. 250 mW was typically used, as it is the maximum heat output for the combination of coil and current source used. Lower heat rates corresponded to reduced independence due to heat diffusing out of the gripper. A 3 mil inner diameter was used for the bore, and a circular approximation of a tetrode was used with a 39 µm diameter. Polyimide and epoxy layers were added based on the coil, wire and via dimensions.

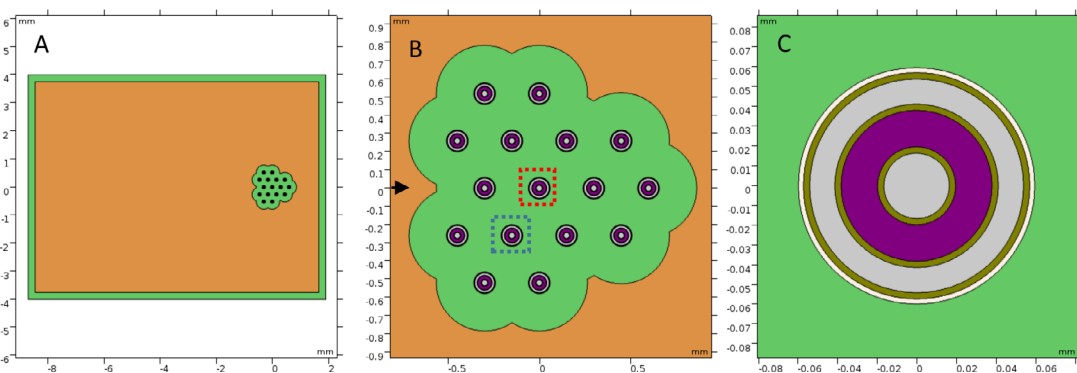

**Appendix 1—figure 3.** Simulation model geometry. (**A**) Gripper board. (**B**) Gripper region with the plotted central gripper and adjacent gripper boxed in red and blue, respectively, and one-dimensional data cutline axis indicated. (**C**) Gripper model.

The material properties were implemented using the material option for copper, as well custom materials for the FR-4, epoxy, coil heaters, PCM, and probes (*Appendix 1—table 1*). In particular, the coil heaters and tetrode probe properties were taken as a volumetric average of the wire metal and polyimide insulation constituents, and the copper plane regions were converted to a blended material that averaged the properties of FR-4 and copper by their approximate proportion in the thickness of the board. The solid-solid and solid-liquid transition for docosane were approximated as a single transition without hysteresis, having a latent heat of 252 J/g and beginning and ending melting temperatures of 41 and 44.5°C. The underfilling of the bore was approximated by then setting the latent heat to 80% of its material value. Temperatures lower than the melting point were assumed to be sufficiently solid and above the highest melting point to be sufficiently liquid. Due to likely imperfections in the model, and non-idealities in the assembly, when assessing sufficiency, an additional safety factor was added (~5°C).

**Appendix 1—table 1.** Material properties of the simulation model.

| Material | Thermal conductivity (W/(m·K)) | Heat capacity (J/(kg·K)) | Density (kg/m$^3$) |
|---|---|---|---|
| FR4 | 0.9 | 1369 | 1900 |
| Copper | 400 | 385 | 8700 |
| FR4-copper blend | 106 | 1111 | 3685 |
| PCM solid | 0.21 | 2480 | 778 |
| PCM liquid | 0.21 | 2760 | 778 |
| Polyimide | 0.15 | 1100 | 1300 |
| Heater wire | 13.1 | 435 | 8110 |
| Coil blend | 8.78 | 657 | 8540 |
| Epoxy | 0.34 | 1184 | 1000 |

Errors in the model are more likely to affect the 'release' than the independence behavior. The gripper release maxima are more dependent on the approximated and idealized geometries within a gripper, which are thin relative to the duration of a heat pulse. The independence behavior is more dependent on the overall gripper geometry, spacing, separating material, and total energy of the heat pulse, which were shorter than the inter-gripper diffusion time. Furthermore, release timing had been previously found for model devices, whereas this simulation mainly investigated a higher gripper count.

Heating periods of prescribed times were applied to sets of the heaters, and the maximum, minimum, and average temperature were monitored in the PCM regions. Starting at a temperature of 20°C, a model gripper could be fully melted after heating at a rate of 250 mW for 12.5 ms. At its

warmest, the minimum temperature of the PCM reached 49.2°C, occurring approximately 3 ms after the heating ended (*Appendix 1—figure 4*). From the beginning of the heating pulse to the time of the maximum temperature, the grippers had practically identical temperature rises and did not have a significant effect on each other; the number of grippers 'on' only changed the maximum by ~1°C. This independence based on the short heat pulses and short distance between a microgripper's heater and PCM relative to the distance between microgrippers greatly simplifies the use of the microdrive. The number of grippers turned on did affect the cooling time for the gripper region over the 100 millisecond time scale, with the effect being greater toward the center of the pattern.

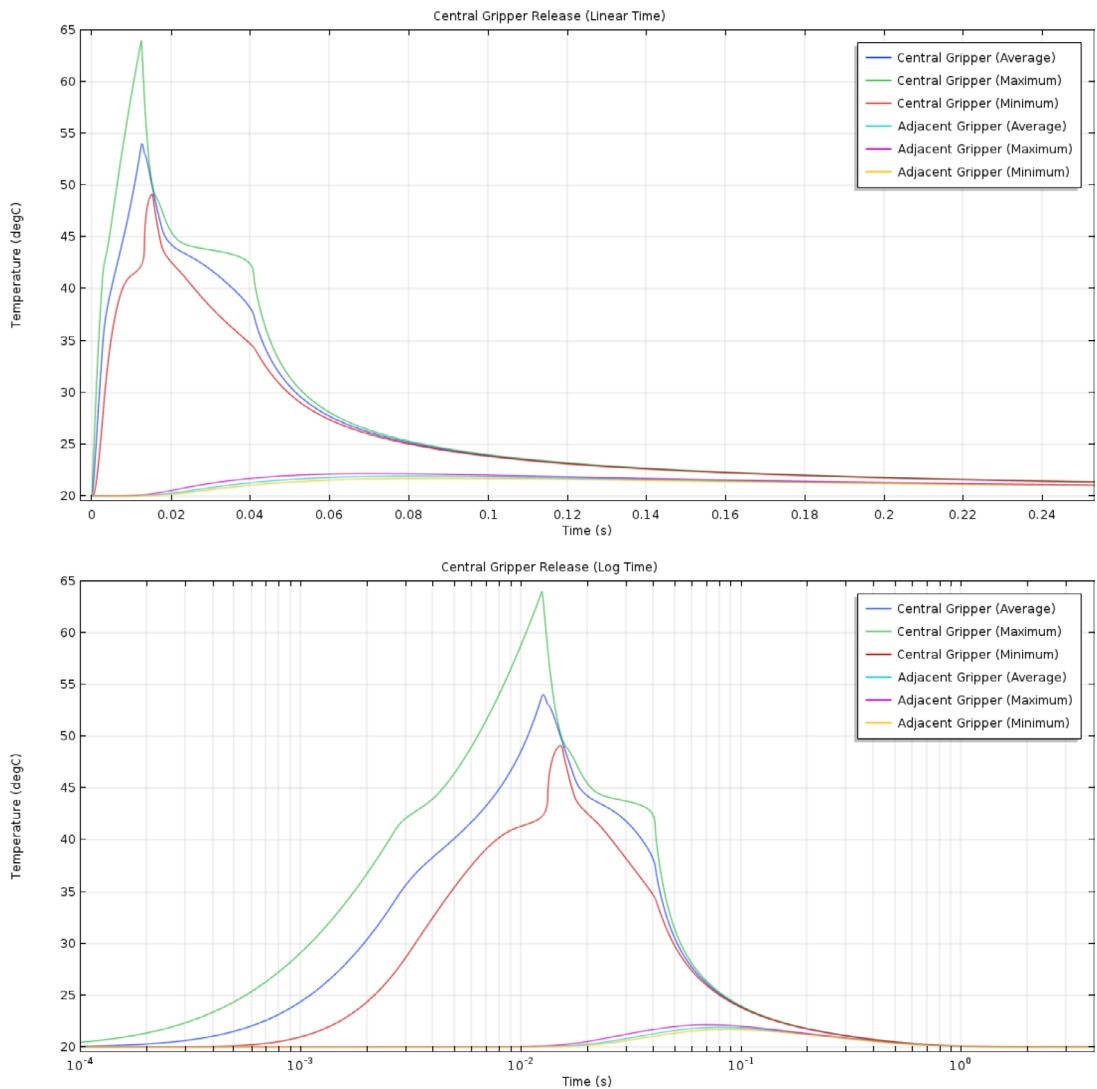

**Appendix 1—figure 4.** (Top) Linear and (bottom) log time plots of the simulated average, maximum, and minimum temperatures for the phase change material in a single 'on' gripper and one adjacent to it.

For independence, the worst-case thermal scenario is when the central gripper is 'off' but all of the surrounding 15 are 'on'. Using the same 12.5 ms heating time, the center gripper reached a maximum temperature of 34.3°C ~110 ms after the beginning of the heat pulse (*Appendix 1— figure 5*, *Appendix 1—Videos 1 and 2*). The temperature change in the central gripper indicates substantial inter-gripper thermal coupling, but the heating efficiency and phase change non-linearity enable independence.

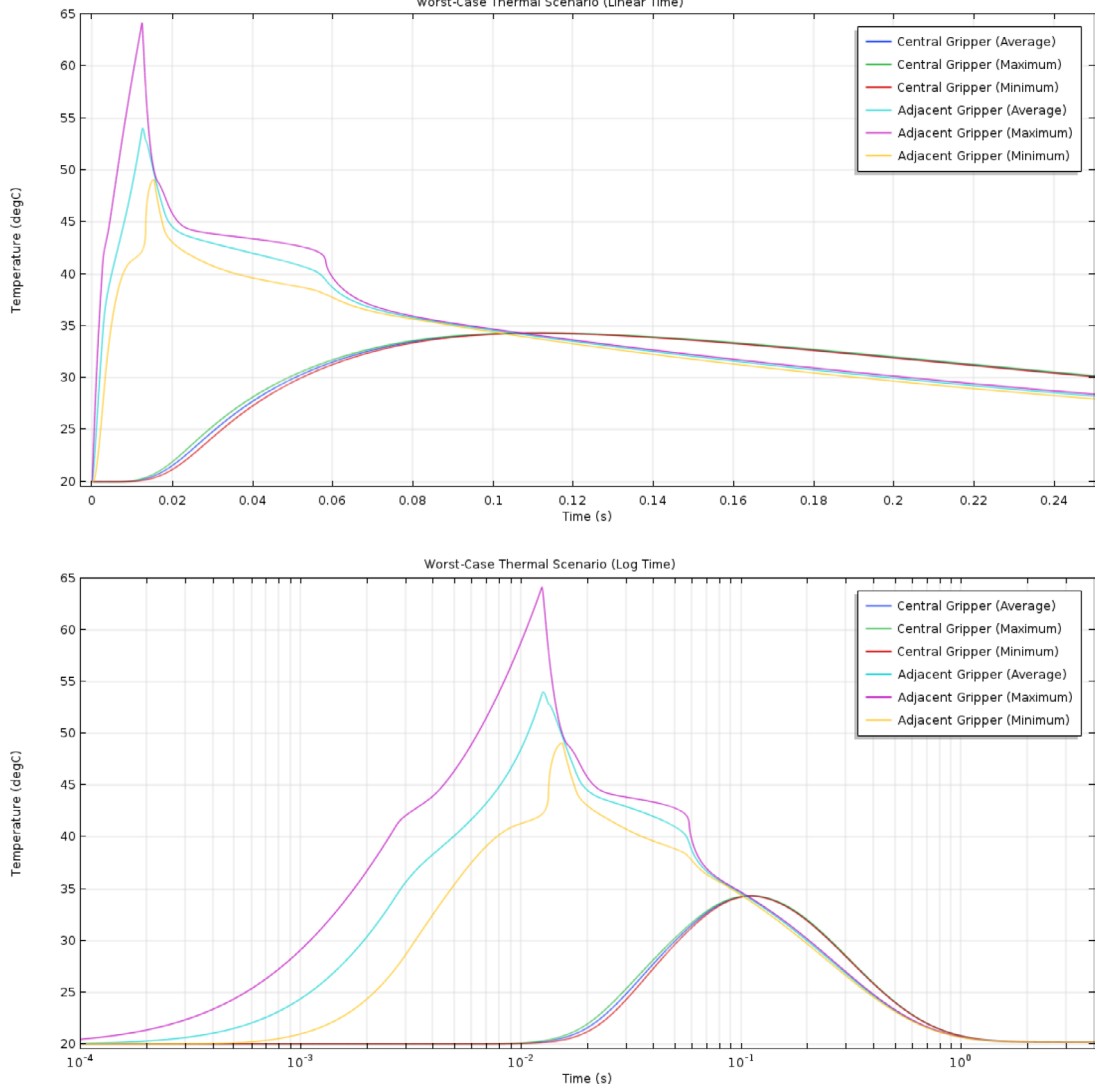

**Appendix 1—figure 5.** (Top) Linear and (bottom) log time plots of the simulated average, maximum, and minimum temperatures for the phase change material of an 'on' gripper and the central 'off' gripper in the 'worst-case' scenario.

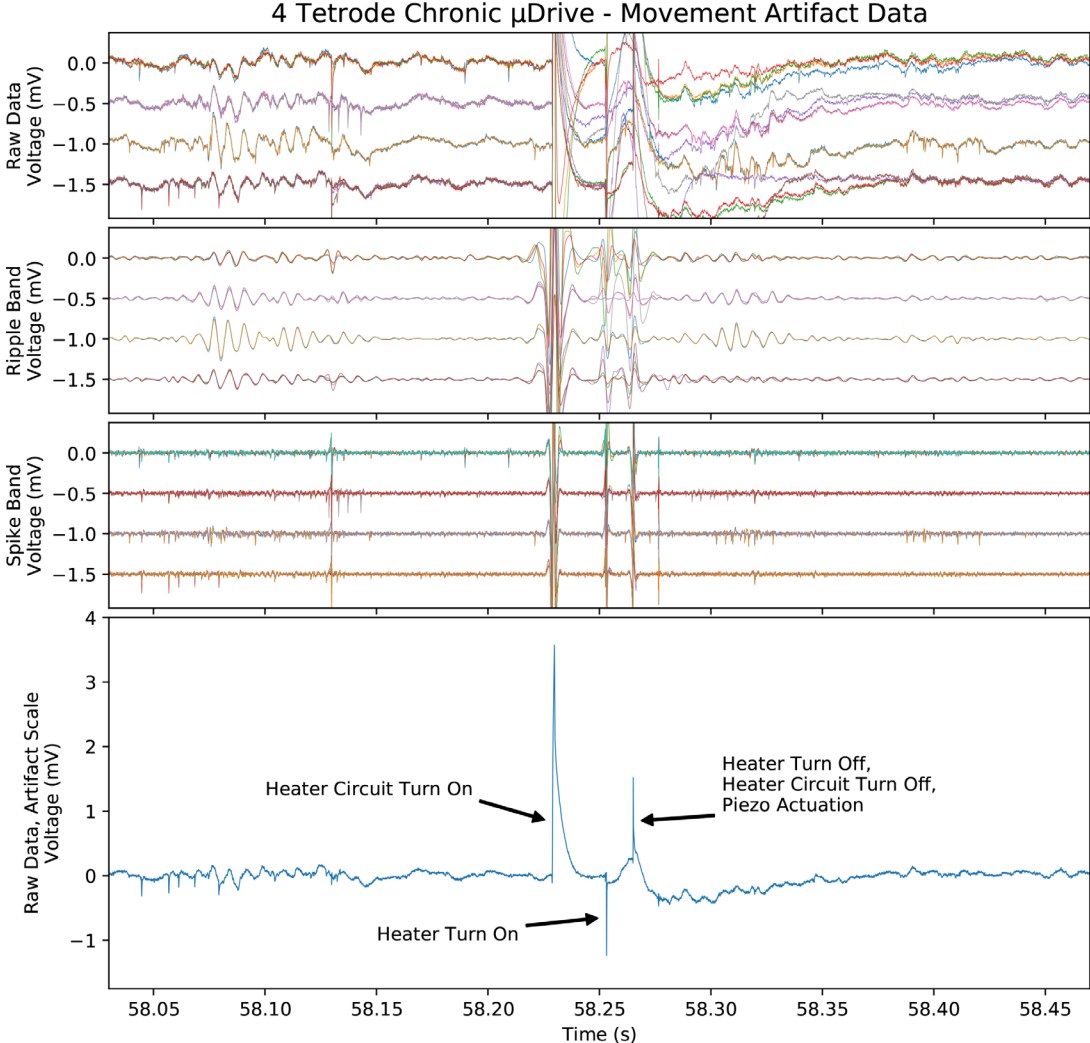

**Appendix 1—figure 6.** Electrical pick-up artifact from one half cycle of probe adjustment during chronic recording. Top three plots, artifact at the scale of and with the same filtering as the chronic data in *Figure 7*. Bottom, unfiltered and full-scale artifact on a single electrode with major sources labeled. Note that the coil voltage was turned off between steps ('heater circuit turn on/off').

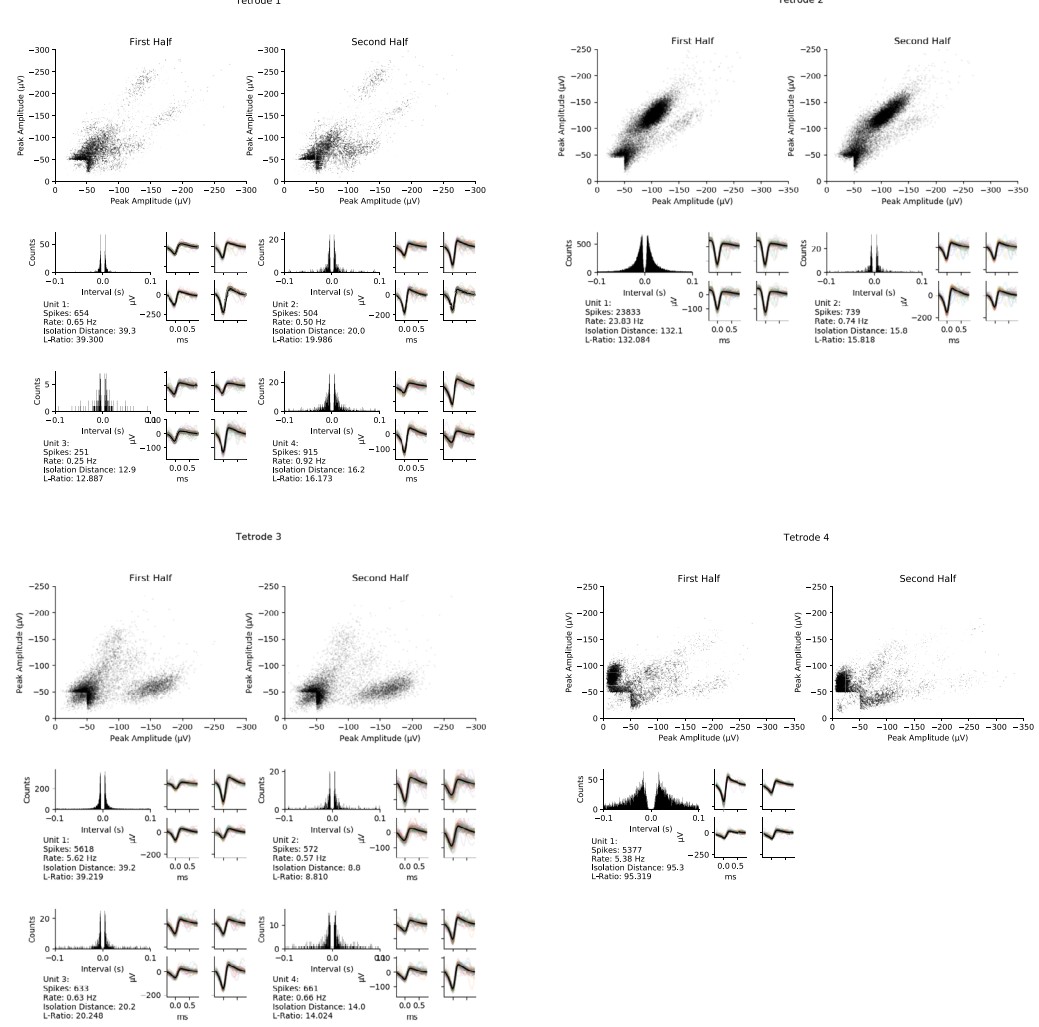

**Appendix 1—figure 7.** Here and next three pages: All isolated units from the four chronically implanted tetrodes (one page per tetrode) during the ~17 min period between successive tetrode adjustments, recorded from an unrestrained, freely behaving rat. Note that some of the data from tetrode 1 is also shown in *Figure 7E–F*. Top, peak amplitude of each spike recorded on two of the four channels of this tetrode during the first and second halves of the period. Note that tetrodes 1 through 3 showed stable activity during the full period while tetrode 4 showed some shifting during the period, limiting the number of isolatable units from that tetrode. Bottom, inter-spike interval histograms and waveforms of the spikes on the four channels for each isolated unit during the full period. Average waveform on each channel (thick black line) and individual spike waveforms (thin colored lines) of 100 evenly spaced spikes of each unit (right), and features of each unit including spike sorting quality metrics isolation distance and L-ratio (lower left).

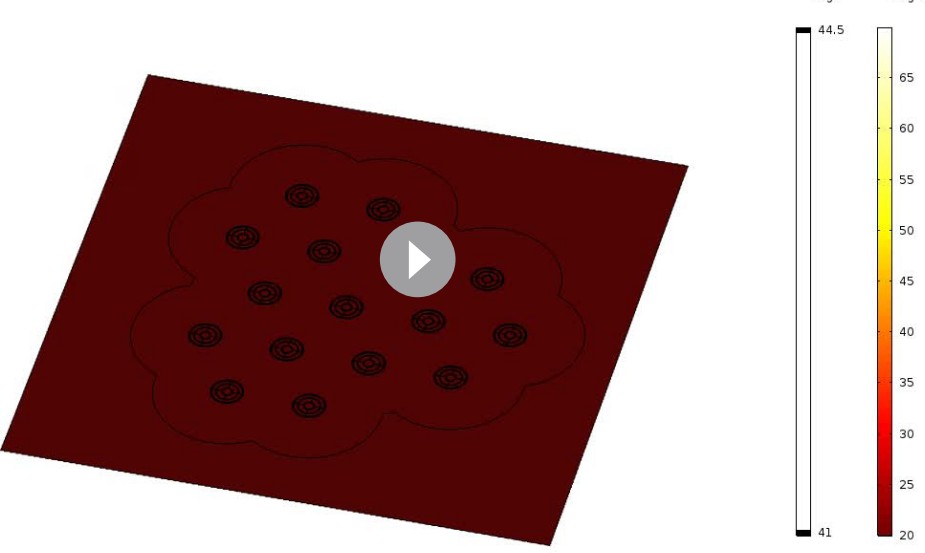

**Appendix 1—video 1.** Gripper board 15-on-1-off thermal simulation.
https://elifesciences.org/articles/71876/figures#video1

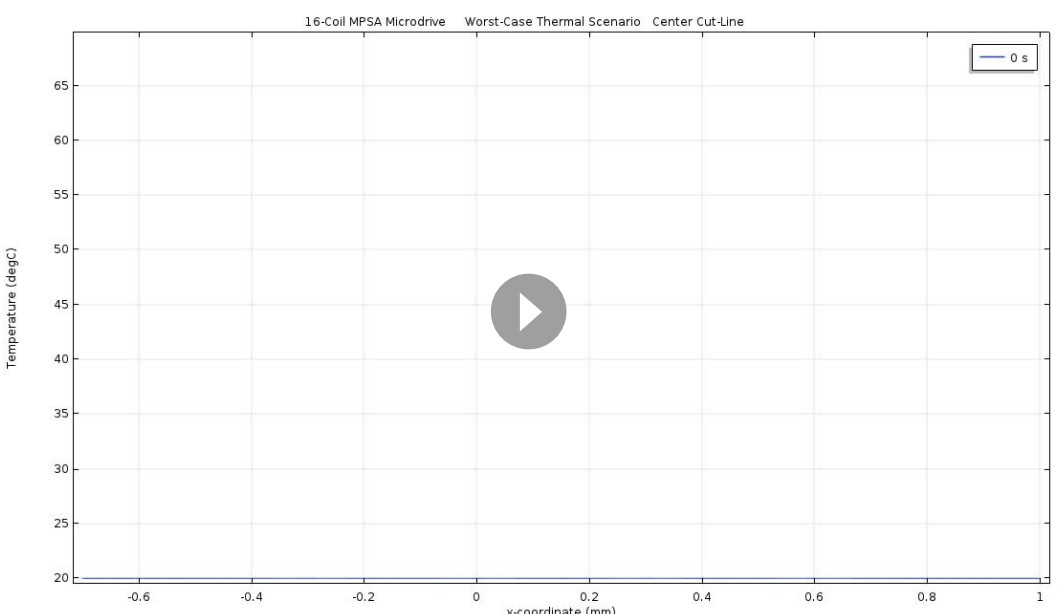

**Appendix 1—video 2.** Center cut-line of 15-on-1-off thermal simulation.https://elifesciences.org/articles/71876/figures#video2

