## [Editor Report]

This work describes a new device for controlling the positioning of chronically-implanted movable electrodes in the brain. Potentially replacing microscrew-based devices with a cunningly-engineered electromechanical system, this highly accurate yet low-cost alternative could be an important milestone for systems neuroscientists currently using microdrives. While the design and demonstration of the system are solid and generated significant excitement, the limited demonstration of the robustness of the device in long term many electrode configurations was perhaps incomplete. On the whole, this promising study suggests that methodological advances may yet revolutionize neuroscience utilizing arrays of movable electrodes.

---

## [Decision Letter]

**Decision letter after peer review:**

Thank you for sending your article entitled "Robotic Multi-Probe-Single-Actuator Inchworm Neural Microdrive" for peer review at *eLife*. Your article is being evaluated by 3 peer reviewers, one of whom is a member of our Board of Reviewing Editors, and the evaluation is being overseen by Laura Colgin as the Senior Editor.

There was consensus among the reviewers that three items seemed to have significant relevance to the potential impact of the work. First, there was concern that the mechanism for gripping the electrodes might damage them (potentially explaining the small number of chronic recording electrodes). Second, there was concern that the heating process might damage the brain due to induced currents or heat transfer. Third, there was concern that the very limited data on cross-talk in electrode motion might neglect important variability in this effect. Finally, as mentioned above, it is our general policy not to ask for significant new experiments. That said, given that the inch-worm drive would most likely find its primary use in chronic experiments, it was disappointing that there was so little chronic data (in terms of analysis and/or animals). This concern is secondary to the previously mentioned 3 items.

Please see the full reviews below:

*Reviewer #1 (Recommendations for the authors):*

Despite some efforts to find alternative approaches, most neuroscience experiments using chronically-implanted movable electrodes rely on screws with a pitch (travel distance for one complete turn) of 200-320 um. A careful experimentalist can probably adjust a screw in tenths of a turn, meaning that the minimal repeatable level of positional change is 20-30 um. Given that this is comparable to the size of the soma of a rodent neuron, precise positioning of electrodes is very challenging. While the advent of nanofabricated arrays of densely spaced electrodes (e.g., the Neuropixel) enables "electrical steering", dozens, if not hundreds of labs around the world still regularly rely on screw-based microdrives using an array of tetrodes or other small-channel-count electrodes.

This work presents a new concept for micropositioning electrodes in chronic neural recording. The "Inchworm Neural Microdrive" uses temperature-dependent phase change materials to grip electrodes at two locations along their axis. By releasing the gripper at the top or bottom, and using a piezo actuator to shift the separation between the grippers with high precision, an electrode can be slowly – but with 5 μm accuracy! – advanced. The authors do an excellent job of describing the manufacturing and characterization of their system for arraying grippers to form a multi-electrode microdrive. They also present preliminary data showing functionality in vivo in both an acute preparation and chronically implanted on a rat. As presented, it is a beautiful exposition of a potentially exciting new technology.

Full enthusiasm is diminished unfortunately by some concerns about usability. In particular, the grippers rely on heating elements in a printed circuit board. It appeared from the design that these formed a coil around the via that might create magnetic fields that would potentially induce currents in the electrodes that could be noisy (making it impossible to evaluate electrode movement using feedback) or even damage the brain tissue. Addressing this concern is probably vital for the future of this technology. There are secondary concerns that are probably less critical. First, one nice thing about a screw is that both rapid and slow advances are possible. In contrast, with the inchworm, it might be so slow to advance the electrodes that maintaining an awake animal quietly in an appropriate behavioral state might be challenging. Second, it was unclear from the preliminary data whether the grippers were damaging the electrodes by removing insulation.

There are 4 concerns outlined in the last paragraph that probably should be addressed. Thoughts about how these could (or could not) be addressed in existing data:

1) Does actuating an electrode induce a current in it or adjacent electrodes that causes damage?

Potential response – if you have histology of heated versus unheated electrodes, that might very easily address this. (The absence of a microlesion being the critical observation.) Of course, localizing electrodes *without* lesioning is challenging, so this may or may not be trivial.

2) Does actuating an electrode induce a current in it or adjacent electrodes that prohibits recording during advancement?

Potential response – if you have this data, please show it. And it's worth commenting about usability – maybe if it is too noisy, you could talk about how long it would take to move an array of electrodes from the corpus callosum to stratum pyramidale, and what fraction of time you would use for feedback versus movement. Also, if its not noisy during movement then there is almost certainly no lesioning current to worry about, I would think?

3) Is 9 μm per step so much slower as to be non usable?

Relatedly, we typically advance our electrodes 3-4 mm in a rat (from above the brain to CA1 or CA3) *after* implant. I know some labs will implant the drive with the electrodes already extended, but I suspect we are not unique. With screws, we typically go a few mm the first day and then more slowly. I think discussing what can be done in 20 minutes or an hour or some limited period, comparing screws and your system, would be ideal. Figure 6 could be easily extended to show how long the timeline *without gaps* is!

4) The single chronic animal had only 4 tetrodes of data.

I know that a new graduate student takes time to learn how to spin and load tetrodes without causing shorts or non-connected channels, but the significant reduction from 12 tetrodes worth of gripper space to 4 tetrodes raised the concern that perhaps 12 were loaded and only 4 worked because, e.g., the gripper damages them. Assuming this is *not* the case, some sort of information about electrode reliability over time or over multiple grip/release cycles would be valuable. Ideally there would be more than one chronically implanted animal, as this, finally, is the use case that the readers care about.

*Reviewer #2 (Recommendations for the authors):*

Smith et al. present a novel and potentially useful new design to improve neural microdrives by using a single electrical actuator to independently control multiple probes of various types. Broadly in neural recordings, electrodes of various types must be placed in a target location in the brain with micron precision. In standard microdrives, either one actuator moves a group of coupled electrodes or each electrode has its own actuator. Both approaches have significant disadvantages. If all electrodes are coupled, the experimenter is severely limited in how they adjust each, leading to some electrodes ending up off-target. If each electrode has an actuator, this adds significant weight and complexity to the drive. In both cases, electrodes are typically moved manually, a laborious process that can stress the animal and lead to microdrive damage because of the many fragile components involved. The authors inchworm approach uses a single piezo to move multiple electrodes by selectively releasing each electrode via heating of a phase-change material that holds the top and bottom of the electrode. The authors characterize the range, accuracy, and independence of electrode movement with this device. They show this approach generalizes to glass electrodes. They also demonstrate successful tetrode recordings from a 16-tetrode acute experiment and a 4-tetrode chronic experiment. Overall, this approach is potentially useful to many using microdrives. Some clarifying questions remain about how generalizable and readily adoptable this approach is and if the resulting recordings are high quality, high yield, and stable.

Overall I found the approach novel and potentially impactful with careful characterization of the device.

– Does this approach generalize to silicone probes, as these are also widely used in the field, especially with the adoption of neuropixels and probes with integrated optical fibers or LEDs? These could fail in this design because they are very brittle and breakable and/or because they have large adapters that would interfere with each other. Even if the authors do not build and test a silicone probe drive, if they could work out whether this is feasible based on their knowledge of the device and typical probes that would go a long way to demonstrating even broader utility and generalizability.

– How readily buildable is the device and readily available are the materials for a typical lab? Is this something a lab would have to buy or build? If long does building take for a novice or expert?

– Brain heating can cause damage at high levels or alter neural activity at low levels. How much heat does the implant generate at the brain surface or at the animal's head, especially when moving many electrodes long distances? Do the electrodes get heated and if so how might that effect their function? Would a cooling system or limits on how long the actuator can run at a time be required?

– The authors carefully characterize accuracy and reliability of forward motion but little is discussed about lateral motion of the electrode while moving. Lateral motion can damage cells and cause inflammation. What is the typical lateral motion of the electrode that is moved?

– The authors demonstrate their inchwork Microdrive in acute and chronic tetrode recordings. Electrophysiologists will want to see as much detail as possible on these recordings to be convinced to adopt this approach. What is the typical cell yield per tetrode and cluster quality using this approach? How stable are the recordings, eg how long is a single cell cluterable? Please show more zoomed in (on x axis) images of recording traces in Figure 6c. Please show single waveforms and clusters in Figure 7d.

– The authors state the 4 tetrode microdrive weighed 4.5 g. How does that compare to a standard 4 tetrode Microdrive? How much does the inchworm actuator add in weight to any standard Microdrive? Weight is especially important for the potential to use this approach in mice, where microdrives must be much lighter than for rats.

– How robust is the system, especially in a chronic implant where the animal may bang the implant often? What are potential failure points and are they fixable in an implanted animal?

*Reviewer #3 (Recommendations for the authors):*

In the current manuscript, Smith et al. describe a system to remotely control the advancement of wires or similarly shaped materials into the brain for in vivo recording. The system combines a single piezo actuator with phase change material on two separate boards to allow multiple wires to be independently controlled via a single actuator, reducing the overall weight of the implant. The system is quite clever and, to my knowledge, novel.

I am very enthusiastic regarding this technology, as I believe it will meaningfully advance tetrode recording quality. Although many labs in the field have moved to silicon probe technology, there are many advantages of tetrodes, which the authors mention in the manuscript. If this method is relatively easy to implement or can be packaged as a purchasable product, I foresee broad adoption of this technology.

The manuscript itself is well-written and the logic and manufacturing process was easy to follow.

I only have a few requests for the authors:

First, I would like to see the repeatability, accuracy, and cross-talk tests repeated on multiple tetrodes. My understanding from the text and the data in Figure 4 is that these values were tested on only a single tetrode (or two tetrodes for the cross-talk test). Specifically, I would like to know whether the inevitable variability in the manufacturing process (different amounts of PCM, differences in the heat coil construction, etc.) contributes to changes in repeatability/accuracy/cross-talk. Basically, I would like the data in Figure 4 to have an N greater than 1 (N=3 would be fine).

Second, I would like some discussion regarding the reusability of the boards. Can a tetrode that has been in the brain (and is now at least partially covered in CSF and blood) be pulled back through the PCM so that the board and PCM can be reused? Does the PCM need to be completely cleaned from the board for the next implant construction? If so, are there methods to do this?

Third, can the authors comment on what they think is a practical limit on the diameter of a cylindrical object that can be moved via this technology? It seems that at a certain size, a cylinder would require so much PCM that it might not be able to all be melted quickly enough. Can this method be used to move 200 um-diameter optic fibers? 400 um-diameter? I'm not asking the authors to rigorously test these different sizes, but rather to give some general guidance based on their experience/knowledge.

---

## [Author Response]

There was consensus among the reviewers that three items seemed to have significant relevance to the potential impact of the work. First, there was concern that the mechanism for gripping the electrodes might damage them (potentially explaining the small number of chronic recording electrodes).

We appreciate this main concern by the editor and the reviewers. However, we can attest that this has never been a problem with our devices over many hours of operation and thousands of steps of gripping and releasing recording electrodes per device. The reason is that, fundamentally, the releasing and gripping of the electrodes by our device gripper during stepping is quite gentle. The method is based on the melting and solidifying of the phase change material around the electrode and inside the PCB via no matter what the shape or material of the electrode is (tetrode, glass capillary, Silicon probe, optical fiber). This mechanism of gripping and releasing of electrodes ensures that there is no large or sudden gripper movement (as would perhaps be expected from other mechanically based grippers) that might potentially damage the electrodes. This was diagrammatically shown in our manuscript Figure 1, but perhaps not properly emphasized in our original manuscript. That has now been corrected and updated in the revised manuscript using the explanations given above.

We also clarify and emphasize the second point regarding our gripping and motion method that we did not appropriately explain in the original manuscript. It is important for us to state that during the motion of the electrodes in our device, as diagrammatically shown in our manuscript Figure 2, the MPSA mechanism does not cause additional axial strain or compression on the electrodes , which would perhaps cause damage if they were. During the electrode motion while stepping, the electrode is only held by one and only one of the electrode grippers (either top of bottom) but never both. Therefore, extension or contraction of our piezo actuator itself during motion never extends or contracts the electrode itself. Furthermore, the electrodes held and positioned by our device are never fully loose while being translated. They are always gripped by at least one gripper at any one time during translation, as Figure 2 of our manuscript describes. This ensures the stability of any one electrode while the adjacent electrodes might be positioned as needed.

We realize now that we did not appropriately emphasize those important points in the original manuscript, and based on the reviewer concerns and our explanations above, we have now updated the revised manuscript clarifying those features of our device.

Second, there was concern that the heating process might damage the brain due to induced currents or heat transfer.

We also appreciate this concern from the Editor and the Reviewers. However, we can again state that the heating of the brain due to operation of our device is essentially negligible. During the robotic stepping of the device for electrode translation described in our manuscript Figure 2, the heating pulses required for melting of the phase change grippers are extremely short (~10ms), while the piezo actuators steps (which do not generate heat) are ~1 second long. Therefore, the heaters in any of the PCBs of our device are on for only up to ~1% of the overall stepping motion time. Additionally, that small amount of heat energy generated by the heaters within the PCB vias is (by design of the copper layers of the PCB) rapidly dissipated within the large PCB volume, and does not thermally conduct to the brain through the very large thermal resistance of the microscopic recording electrodes. Furthermore, the phase change material for this device was chosen so that the grippers would not need to rise to a temperature significantly above the typical temperature of the brain. Finally, we add that during the electrode recordings from neurons, or simply during the “hibernate state” of the device, all the electrodes are fixed (gripped) in the device, which by definition occurs when all the heaters in the device are off and generate no heat. We realize that originally we have not properly addressed these heating concerns in our device, but have now added the above points to the revised manuscript.

Third, there was concern that the very limited data on cross-talk in electrode motion might neglect important variability in this effect.

We thank the Editor and the Reviewers for bringing up this concern. We break down the issue of cross-talk into electrical and mechanical components.

Regarding electrical cross-talk, there is most certainly electrical pick-up cross-talk during electrode motion, as is to be expected from repeatedly and quickly powering the ~10 V heating coil voltage and applying the ~50 mA electrical currents through the micro-heaters in the PCB vias that surround the recording electrodes as well as applying the 100 V piezo actuation steps. However, the electrical recordings from neurons with sensitive neural amplifiers coupled to the electrodes were regularly performed during the stepping motion of the device without apparent issue to the tissue, electronics, or recording quality (as seen in the chronic recording data in Figure 7 that was taken after thousands of steps). It was never necessary to disconnect or power down the elements of the device that perform the motion steps (piezo actuator and the heaters) to improve the data quality during recording, but if motion was not being performed during a recording session, it was also not necessary to use or power these sections. Therefore, the described MPSA mechanism in no way affects the ability to perform effective neural recordings. We have now included this clarification in the revised manuscript. We have also submitted the new Appendix 1-figure 6 to the updated manuscript that addresses this topic in detail and illustrates this particular feature of concern in the device.

Regarding physical cross-talk (i.e. the extent to which the stepping movement of one electrode could cause unintended movement of other electrodes), we have added new analysis of such movements of the other electrodes in the on-axis (axial) as well as lateral (off-axis) directions, both in tests in the air and in agar (simulating what happens in the brain). We have added individual trial data in Figure 4F. Together, the data show that there is very little unintended movement of the other electrodes in an agar brain phantom. Again, the quality of the chronic recording data (described below) show that in the real, chronically implanted animal brain, any small movements do not degrade the ability to obtain high-quality single-unit spiking data matching that obtained with standard, manually adjusted drives.

Finally, as mentioned above, it is our general policy not to ask for significant new experiments. That said, given that the inch-worm drive would most likely find its primary use in chronic experiments, it was disappointing that there was so little chronic data (in terms of analysis and/or animals). This concern is secondary to the previously mentioned 3 items.

We appreciate the reviewers’ concern. This was fundamentally a logistics problem due to staff changes and transitions of the first, second, and last authors to different institutions during the course of this research project, which prevented us from doing more. We felt strongly that our work is particularly suited for the *eLife* Tools and Resources article category where we do not report major new biological insights or mechanisms, but where it is clear from our extensive testing and characterization experiments, and from our proof-of-concept in vivo experiments, that our novel robotic device will enable such advances to take place. We would greatly appreciate understanding from the Editor and the Reviewers on this point listed as a secondary concern. In the revision, we did what was possible, by performing significant additional analysis of the chronic data in terms of cluster separation, the number of detected units, and the quality and stability of our recordings in a freely behaving animal taken several weeks after drive implantation. The results of this analysis are shown in the new panels Figure 7E-F and in the new Appendix 1-figure 7, and are also described in the revised text as well as methods section. These results demonstrate that the chronic recordings using this drive are of the same quality as those using standard manually adjusted.

Reviewer #1 (Recommendations for the authors):Despite some efforts to find alternative approaches, most neuroscience experiments using chronically-implanted movable electrodes rely on screws with a pitch (travel distance for one complete turn) of 200-320 um. A careful experimentalist can probably adjust a screw in tenths of a turn, meaning that the minimal repeatable level of positional change is 20-30 um. Given that this is comparable to the size of the soma of a rodent neuron, precise positioning of electrodes is very challenging. While the advent of nanofabricated arrays of densely spaced electrodes (e.g., the Neuropixel) enables "electrical steering", dozens, if not hundreds of labs around the world still regularly rely on screw-based microdrives using an array of tetrodes or other small-channel-count electrodes.This work presents a new concept for micropositioning electrodes in chronic neural recording. The "Inchworm Neural Microdrive" uses temperature-dependent phase change materials to grip electrodes at two locations along their axis. By releasing the gripper at the top or bottom, and using a piezo actuator to shift the separation between the grippers with high precision, an electrode can be slowly – but with 5 μm accuracy! – advanced. The authors do an excellent job of describing the manufacturing and characterization of their system for arraying grippers to form a multi-electrode microdrive. They also present preliminary data showing functionality in vivo in both an acute preparation and chronically implanted on a rat. As presented, it is a beautiful exposition of a potentially exciting new technology.Full enthusiasm is diminished unfortunately by some concerns about usability. In particular, the grippers rely on heating elements in a printed circuit board. It appeared from the design that these formed a coil around the via that might create magnetic fields that would potentially induce currents in the electrodes that could be noisy (making it impossible to evaluate electrode movement using feedback) or even damage the brain tissue. Addressing this concern is probably vital for the future of this technology.

We thank Reviewer #1 for bringing up this important issue. We have addressed this topic in the overall response to Main Concern #3 above and believe that we have provided sufficient explanation and data to alleviate concerns over this issue. In particular, the steps are very brief and the recording system is left on during each step, and therefore the effect of a step can be evaluated immediately after the step and used for rapid feedback, as can be seen in the new Appendix 1-figure 6. We cannot easily evaluate the effect of the step during the ~10 ms constituent pulses themselves, but right after. Feedback on this timescale will be rapid enough for future applications, such as automated adjusting. Furthermore, regarding the question of potential tissue damage, after many steps through the hippocampus we are still able to record high-quality single unit data in a chronically implanted animal (shown in the new panels Figure 7E-F and the new Appendix 1-figure 7).

There are secondary concerns that are probably less critical. First, one nice thing about a screw is that both rapid and slow advances are possible. In contrast, with the inchworm, it might be so slow to advance the electrodes that maintaining an awake animal quietly in an appropriate behavioral state might be challenging.

It is true that the maximum translation speed for any probe is set by the rate at which the grippers can be cycled. However, we emphasize that our device has a continuous cycling speed of 0.5 Hz (and burst speed of 2.5 Hz), i.e. ~5 microns per second (~9.5 microns per step) yielding ~1 mm of electrode translation in ~3 minutes. In our experience, this is sufficiently fast for common neural microdrive usage with laboratory animals (e.g. for targeting the hippocampus in chronically implanted rodents). This is certainly the case for the medium and fine adjustment in chronic animals. For the commonly rapid initial advance of ~1 mm (right after implantation) when the target is, say, the hippocampus, the ~3 minutes needed per tetrode is fast enough (as demonstrated in our chronic animal), especially if one advances say 2 non-adjacent tetrodes at a time (which is not possible with manual adjusting), which would allow 16 tetrodes to be advanced to ~1 mm depth in ~24 minutes. Also, for medium- and fine-scale adjustments (e.g. on the approach from the callosum to the hippocampal cell layer), we generally prefer to adjust electrodes when the animal is resting (whether awake or asleep). The remote-control aspect of our drive means we do not need to hold the drive and/or animal, nor touch the drive with the screwdriver, for each set of turns for each tetrode. This is practically very useful because it is less likely the animal will suddenly become very active, which otherwise slows down the adjusting process itself, as well as slows down the rate of receiving steady-state feedback on the result of each adjustment because of large neural activity changes due to sudden behavioral activity. We now clearly state these adjusting parameters for consideration by potential users in the revised manuscript.

Second, it was unclear from the preliminary data whether the grippers were damaging the electrodes by removing insulation.

We thank Reviewer #1 for bringing up this issue, which was designated as Main Concern #1 in the overall evaluation, and which we addressed there. In addition, the temperature to melt the low temperature PCM is far below the melting or degradation temperature of the electrode insulation or bond coat.

There are 4 concerns outlined in the last paragraph that probably should be addressed. Thoughts about how these could (or could not) be addressed in existing data:1) Does actuating an electrode induce a current in it or adjacent electrodes that causes damage?Potential response – if you have histology of heated versus unheated electrodes, that might very easily address this. (The absence of a microlesion being the critical observation.) Of course, localizing electrodes *without* lesioning is challenging, so this may or may not be trivial.

Unfortunately, we were unable to acquire this histology data. However, as this was one aspect of the main concerns listed in the overall evaluation, we have addressed it above to the best of our ability. Furthermore, as mentioned above, the quality of the chronic recording data, which we have now analyzed in more detail for the revision (Figure 7E-F and Appendix 1-figure 7), suggests that there was minimal damage, if any.

2) Does actuating an electrode induce a current in it or adjacent electrodes that prohibits recording during advancement?Potential response – if you have this data, please show it. And it's worth commenting about usability – maybe if it is too noisy, you could talk about how long it would take to move an array of electrodes from the corpus callosum to stratum pyramidale, and what fraction of time you would use for feedback versus movement. Also, if its not noisy during movement then there is almost certainly no lesioning current to worry about, I would think?

We have addressed this question in detail above, and have provided additional data as requested in the new Appendix 1-figure 6. To recap, this artifact is much shorter than the time a researcher would typically wait to see the effect of movement on the recording, and is therefore not a practical hindrance (i.e. a few tens of milliseconds are lost to noise per ~10 microns advanced, which totals <10 seconds of time lost to noise artifacts while traversing the ~300 micron distance from the corpus callosum to the stratum pyramidale, which is usually done over a few days according to standard adjusting strategies).

3) Is 9 μm per step so much slower as to be non usable?Relatedly, we typically advance our electrodes 3-4 mm in a rat (from above the brain to CA1 or CA3) *after* implant. I know some labs will implant the drive with the electrodes already extended, but I suspect we are not unique. With screws, we typically go a few mm the first day and then more slowly. I think discussing what can be done in 20 minutes or an hour or some limited period, comparing screws and your system, would be ideal. Figure 6 could be easily extended to show how long the timeline *without gaps* is!

We addressed this question above, but restate here with more details. In our device we used a continuous cycling speed of 0.5 Hz, i.e. one actuation step of ~10 microns per two seconds, which results in ~1 mm of maximum electrode translation in ~3 minutes. (For short periods, it is possible to cycle at 2.5 Hz, yielding ~1.5 mm of translation in 1 minute.) At the standard ~10 micron steps at 0.5 Hz, this would allow us to lower 16 tetrodes, one at a time, to ~1 mm depth in ~55 minutes. However, one of the benefits of our device is that multiple probes can be moved in at the same time, leading to multiplicative reductions in total translation time depending on the size of the groups moved. For instance, we could lower 2 non-adjacent tetrodes at a time (to avoid a “bed of nails” effect) and thus lower all 16 tetrodes by ~1 mm in under half an hour. In general, with larger numbers of tetrodes, one could lower more tetrodes simultaneously while minimizing “bed of nails” effects (since with more tetrodes, they would necessarily be spread out over a larger area). In Videos 2-6, we demonstrate the ability to either move probes individually or in groups. Further, the operator could control and script these movements remotely. Regarding the timeline for Figure 6C, we have added the total time we spent to go from 0 to 16 tuned tetrodes, which was ~1 h 10 min (though we were not optimizing for speed in this experiment).

4) The single chronic animal had only 4 tetrodes of data.I know that a new graduate student takes time to learn how to spin and load tetrodes without causing shorts or non-connected channels, but the significant reduction from 12 tetrodes worth of gripper space to 4 tetrodes raised the concern that perhaps 12 were loaded and only 4 worked because, e.g., the gripper damages them. Assuming this is *not* the case, some sort of information about electrode reliability over time or over multiple grip/release cycles would be valuable. Ideally there would be more than one chronically implanted animal, as this, finally, is the use case that the readers care about.

We thank the reviewers for this point, which was listed as an overall but secondary concern in the consensus review. We have responded to this inquiry in the comments above. We confirm that it was definitely not the case that 12 tetrodes were loaded and only 4 worked. Four tetrodes were loaded into the chronically implanted drive and all 4 tetrodes worked as intended. There were no issues in electrode reliability over time and, as we stated previously, the electrodes were gripped/released over thousands of cycles without problems. The new Appendix 1-figure 7 shows that all 4 tetrodes made it to the hippocampal pyramidal cell layer, and all showed clusters of similar quality to what is seen with standard manually operated drives (noting that one tetrode had some cluster drift, presumably due to the fact that the tetrodes were stepped immediately before the recording period). Aside from the added complexity of loading additional probes, we do not expect any major hindrances to that scaled up experiment working as well. The drive was loaded and operated successfully using 16 tetrodes in the acute in vivo experiment shown in Figure 6, with all 16 tetrodes showing hippocampal spiking. We did not load all 16 tetrodes for chronic testing in order to save time (as the vast majority of the work for this project involved designing and testing the inchworm drive components), and for chronic testing we focused on the proof-of-concept experiment of targeting the hippocampal cell layer in an unrestrained, freely behaving animal.

Reviewer #2 (Recommendations for the authors):Smith et al. present a novel and potentially useful new design to improve neural microdrives by using a single electrical actuator to independently control multiple probes of various types. Broadly in neural recordings, electrodes of various types must be placed in a target location in the brain with micron precision. In standard microdrives, either one actuator moves a group of coupled electrodes or each electrode has its own actuator. Both approaches have significant disadvantages. If all electrodes are coupled, the experimenter is severely limited in how they adjust each, leading to some electrodes ending up off-target. If each electrode has an actuator, this adds significant weight and complexity to the drive. In both cases, electrodes are typically moved manually, a laborious process that can stress the animal and lead to microdrive damage because of the many fragile components involved. The authors inchworm approach uses a single piezo to move multiple electrodes by selectively releasing each electrode via heating of a phase-change material that holds the top and bottom of the electrode. The authors characterize the range, accuracy, and independence of electrode movement with this device. They show this approach generalizes to glass electrodes. They also demonstrate successful tetrode recordings from a 16-tetrode acute experiment and a 4-tetrode chronic experiment. Overall, this approach is potentially useful to many using microdrives. Some clarifying questions remain about how generalizable and readily adoptable this approach is and if the resulting recordings are high quality, high yield, and stable.Overall I found the approach novel and potentially impactful with careful characterization of the device.– Does this approach generalize to silicone probes, as these are also widely used in the field, especially with the adoption of neuropixels and probes with integrated optical fibers or LEDs? These could fail in this design because they are very brittle and breakable and/or because they have large adapters that would interfere with each other. Even if the authors do not build and test a silicone probe drive, if they could work out whether this is feasible based on their knowledge of the device and typical probes that would go a long way to demonstrating even broader utility and generalizability.

We thank the reviewer for this comment, which is part of Main Concern #1, which we have addressed above. To recap, yes, we believe this approach generalizes to many other types of rigid, rod-like probes since the method of gripping and actuation is gentle, relies on phase change material melting and solidification around any cross sectional shape of the probe being gripped, and does not axially stretch or compress the probe. As pointed out, we demonstrated this generalizability by applying the technique to two types of probes, wire tetrodes and glass micropipettes, but any other material and shape is in principle possible to grip and move, such as silicon probes and optical fibers. In practice, regarding silicon probes, the allowable geometry with which the probes can be packed in the devices would be limited by the part of the probe above the shank. Tetrodes do not require any such “base” that is rigidly attached above each electrode, but some probes, such as Neuropixels, do. Our device would be capable of translating Neuropixels probes by holding the shank with the phase change material, but the base would add additional constraints on packing density/geometry.

– How readily buildable is the device and readily available are the materials for a typical lab? Is this something a lab would have to buy or build? If long does building take for a novice or expert?

We believe that any laboratory that is capable of loading conventional screw-based tetrode neural microdrives would be capable of loading probes into an MPSA microdrive. Building the core MPSA motor – the grippers, piezo and board assembly – and developing the control software is in principle possible for a typical systems neuroscience-focused laboratory. The materials are readily available, the techniques are not prohibitively difficult, but the process is specialized. In our experience in a research and development setting, construction of the entire device takes one week of labor for a trained person. However, once the MPSA motor is constructed, it can be reused. Loading of the robotic device with tetrodes is also similar in effort and time to loading of the classic manual tetrode drives. For mass distribution, our opinion is that it makes the most sense to sell the core MPSA motor through a neuroscience tools vendor and provide guidance on probe loading and access to CAD files and software. This work in particular focuses on demonstrating the technique, whereas future work would focus on creating devices that best fit user applications and simplify access. We hope that our proof-of-concept experiments will enable such advances to take place.

– Brain heating can cause damage at high levels or alter neural activity at low levels. How much heat does the implant generate at the brain surface or at the animal's head, especially when moving many electrodes long distances? Do the electrodes get heated and if so how might that effect their function? Would a cooling system or limits on how long the actuator can run at a time be required?

We thank the reviewer for bringing up this issue, which was designated as Main Concern #2 in the overall evaluation. We have addressed this topic in the overall response. To recap, the heat from the grippers does not substantially flow to the brain through the probes. The probes are too thin and long to significantly conduct the heat of the brief pulse to the brain, and instead the heat is distributed throughout the gripper boards and then to the overall external structure of the device. Further, the phase change material for this device was chosen so that the grippers would not need to rise to a temperature significantly above the typical temperature of the brain.

– The authors carefully characterize accuracy and reliability of forward motion but little is discussed about lateral motion of the electrode while moving. Lateral motion can damage cells and cause inflammation. What is the typical lateral motion of the electrode that is moved?

We appreciate this potential concern raised by Reviewer #2. In Figure 4E-G, we performed the requested measurement. We measured the lateral movement (denoted as “y axis” movement in Figure 4E-G) of the tetrodes both in air and in agar (simulating the in-brain mechanical environment). We note in particular that in agar the lateral motion is essentially completely eliminated (Figure 4F). We have updated Figure 4 of the revised manuscript on the mechanical characterization of the microdrive by adding more individual test trials (Figure 4F), which shows the lateral movements in air and the virtual lack of any lateral movements in agar.

– The authors demonstrate their inchwork Microdrive in acute and chronic tetrode recordings. Electrophysiologists will want to see as much detail as possible on these recordings to be convinced to adopt this approach. What is the typical cell yield per tetrode and cluster quality using this approach? How stable are the recordings, eg how long is a single cell cluterable? Please show more zoomed in (on x axis) images of recording traces in Figure 6c. Please show single waveforms and clusters in Figure 7d.

In testing our device in vivo, we did not specifically attempt to optimize cell yield or stability. Instead, we continuously stepped the tetrodes through the CA1 pyramidal cell layer, observing the spike amplitudes on the continuous traces, and generally did not pause for extended periods during this stepping. However, in order to address this question and investigate cell yield and stability in a freely moving animal, we have analyzed a period of our neural recordings when there was a longer gap between successive tetrode adjustments. This period was ~17 min long and we found that the spike amplitudes and waveforms were quite stable and allowed us to isolate multiple single units per tetrode on average (shown in the new panels Figure 7E-F as well as the new Appendix 1-figure 7).

As can be seen, the spike amplitudes on the channels of each tetrode were very stable for 3 of the 4 tetrodes, while some shifting was observable on one tetrode (tetrode 4) – which is not unexpected because this period was immediately preceded (and followed) by multiple adjustments of the tetrode depths by experimenter-controlled steps. The tetrodes showed clearly separable clusters similar to what is seen with standard, manually adjusted screw-based drives. While the yield of 11 units from the 4 tetrodes in this chronically implanted animal is not particularly high, the stability and cluster quality we observed (according to both visual and quantitative cluster quality metrics – spike waveforms, ISI histogram, Isolation Distance, and L-Ratio, as shown in Appendix 1-figure 7 for each cluster) across this period were high, and similar to what is observed with standard, manually adjusted drives. As we did not optimize for cell yield or stability, this result reassures us that the yield and stability would be as good as with such drives.

Regarding Figure 6, we have added a zoomed in image of the traces from Figure 6C in that figure panel.

– The authors state the 4 tetrode microdrive weighed 4.5 g. How does that compare to a standard 4 tetrode Microdrive? How much does the inchworm actuator add in weight to any standard Microdrive? Weight is especially important for the potential to use this approach in mice, where microdrives must be much lighter than for rats.

The proof-of- concept device was designed for up to 16 probes and using lower numbers could allow for shrinking the design. To clarify, the demonstrated 4.5 g chronic 4-probe drive would also have been ~4.5 g if loaded with 16 probes, as the only differences in the mechanical structure are the tetrodes themselves, their pins, and the coil grippers – all of which add negligible weight. This is now clearly stated in the revised manuscript.

Specifically, for a 4-probe mouse device, we would reduce the size of the boards and the cage, as is normally done with other microdrive devices when translated between rat and mouse models. We could also reduce the overall size by using a proportionally shorter piezo for the smaller brain size of a mouse. An expected reduction in the weight by a factor of 2X to 3X is reasonable.

– How robust is the system, especially in a chronic implant where the animal may bang the implant often? What are potential failure points and are they fixable in an implanted animal?

We designed and constructed the external structure of the full microdrive to be robust to chronic rat head strikes, and the ~6-week duration on the animal’s head before the chronic recording data shown in Figure 7 demonstrates that the full device is able to survive such an environment. Without this structure the thin probes are liable to damage due to external exposure as in other devices. The piezo motor is a ceramic block, which could be broken by direct banging to the core MPSA motor. Regarding other potential failure points, it is possible that too much retraction of the implanted probes could bring Vaseline from the cannula into the lower gripper board, which would prevent the gripper from solidifying. We therefore left a ~3 mm gap between these and limited the retraction from the maximum depth. Finally, it is possible that enough PCM could be ejected from the grippers as to eventually prevent a solid grip, but PCM can be brought back in by using a particular heating cycle. We discuss these points extensively now in the updated Appendix document.

Reviewer #3 (Recommendations for the authors):In the current manuscript, Smith et al. describe a system to remotely control the advancement of wires or similarly shaped materials into the brain for in vivo recording. The system combines a single piezo actuator with phase change material on two separate boards to allow multiple wires to be independently controlled via a single actuator, reducing the overall weight of the implant. The system is quite clever and, to my knowledge, novel.I am very enthusiastic regarding this technology, as I believe it will meaningfully advance tetrode recording quality. Although many labs in the field have moved to silicon probe technology, there are many advantages of tetrodes, which the authors mention in the manuscript. If this method is relatively easy to implement or can be packaged as a purchasable product, I foresee broad adoption of this technology.The manuscript itself is well-written and the logic and manufacturing process was easy to follow.I only have a few requests for the authors:First, I would like to see the repeatability, accuracy, and cross-talk tests repeated on multiple tetrodes. My understanding from the text and the data in Figure 4 is that these values were tested on only a single tetrode (or two tetrodes for the cross-talk test). Specifically, I would like to know whether the inevitable variability in the manufacturing process (different amounts of PCM, differences in the heat coil construction, etc.) contributes to changes in repeatability/accuracy/cross-talk. Basically, I would like the data in Figure 4 to have an N greater than 1 (N=3 would be fine).

We thank the reviewer for the comment and have addressed the question in several ways. First, we added clarification in the text that each panel in Figure 4 was performed on separate sets of probes. Second, we added a bottom panel to Figure 4B as an illustration of another probe (a pipette) moving in the microdrive and added traces to Figure 4F to show all 5 steps of the probe. We also clarified in the text that Video 3 (a separate experiment) corroborated the results of the cross-talk data in Figure 4F:

This experiment was also repeated with 3 tetrodes moving in agar where the maximum observed off-axis deflection of the stationary electrode across >100 steps was ~6 µm and the on-axis deflection was negligible (Video 3).

We hope that between the characterization in Figure 4 and the 3 videos that show movement consistent with this characterization, the reviewer will find sufficient evidence that the manufacturing process does not severely impact performance.

Second, I would like some discussion regarding the reusability of the boards. Can a tetrode that has been in the brain (and is now at least partially covered in CSF and blood) be pulled back through the PCM so that the board and PCM can be reused? Does the PCM need to be completely cleaned from the board for the next implant construction? If so, are there methods to do this?

We thank the reviewer for these questions. Yes, our devices are reusable. More specifically, the core MPSA motor and EIC Board are both reusable between implants. The external 3D printed cage is not reusable due to the acrylic adhesive used to secure it to the skull but can be remade fairly easily and inexpensively. The probes themselves are not damaged by the motor but would need to be cleaned before retraction through the boards, which is a process we had performed for acute studies of the device. But generally, as is common practice in manual tetrode drives, new tetrodes are loaded for new chronic experiments. Our practice was to clean out all the PCM between uses as it makes loading new probes through the gripper boards easier and would prevent PCM from getting onto the probe tips. We removed any remaining PCM by heating the boards above the PCM melting point and then blowing air from a compressed air canister through the gripper bores and then running a fine thread through them to ensure proper clearance.

Third, can the authors comment on what they think is a practical limit on the diameter of a cylindrical object that can be moved via this technology? It seems that at a certain size, a cylinder would require so much PCM that it might not be able to all be melted quickly enough. Can this method be used to move 200 um-diameter optic fibers? 400 um-diameter? I'm not asking the authors to rigorously test these different sizes, but rather to give some general guidance based on their experience/knowledge.

We thank the reviewer for these questions. We focused this work on the smaller diameter neural probes, namely the twisted wire tetrode. For any of these larger diameter probes, minimizing the amount of PCM within the via between the probe and the helical heater coil would indeed be important. This would not only reduce the required heat to melt the PCM, but also reduce the lateral motion of the probe during translation, and maintain the capillary action that keeps the PCB within the via. For these reasons, the heater coil would always be carefully sized to satisfy those requirements. With larger probes, there becomes an additional consideration of the amount of heat the probe itself can conduct away from the gripper. This is based on the probe’s cross-section and thermal diffusivity characteristics, where a very thermally diffusive probe (e.g. a 1-mm metal needle) might require modifications to the process in order to fully melt the PCM in the via. While glass has a higher thermal conductivity than the insulating polymer of a tetrode, it is still low relative to metal. Therefore, moving a 200 or 400-micron optical fiber would likely work similarly well to the micropipettes we demonstrated. The key idea remains to use a short heat impulse so that the heat minimally diffuses into the probe relative to the PCM during translation. One possibility is to slightly modify thermally conductive probes by placing a less thermally diffusive sheath around them that would instead be in contact with and delays heat loss from the PCM. Alternatively, and more simply, the heating power to melt the PCM could be increased if there is no risk of melting other nearby grippers. We have added these considerations to the revised Discussion.